# Lipogenic enzyme FASN promotes mutant p53 accumulation and gain-of-function through palmitoylation

Juan Liu[1], Yiyun Shen[1], Jie Liu[1], Dandan Xu[1], Chun-Yuan Chang[1], Jianming Wang[1], Jason Zhou[1], Bruce G. Haffty[1], Lanjing Zhang[2,3], Jill Bargonetti[4], Subhajyoti De[5], Wenwei Hu[1] ✉ & Zhaohui Feng[1] ✉

The tumor-suppressive function of p53 is frequently disrupted by mutations in cancers. Missense mutant p53 (mutp53) protein often stabilizes and accumulates to high levels in cancers to promote tumorigenesis through the gain-of-function (GOF) mechanism. Currently, the mechanism of mutp53 accumulation and GOF is incompletely understood. Here, we identify the lipogenic enzyme FASN as an important regulator of mutp53 accumulation and GOF. FASN interacts with mutp53 to enhance mutp53 palmitoylation, which inhibits mutp53 ubiquitination to promote mutp53 accumulation and GOF. Blocking FASN genetically or by small-molecule inhibitors suppresses mutp53 palmitoylation to inhibit mutp53 accumulation, which in turn inhibits the growth of mutp53 tumors in orthotopic and subcutaneous xenograft tumor models and transgenic mice, as well as the growth of human tumor organoids carrying mutp53. Our results reveal that mutp53 palmitoylation is an important mechanism underlying mutp53 accumulation and GOF, and targeting FASN is a potential therapeutic strategy for cancers carrying mutp53.

p53 plays a key role in tumor suppression. As a transcription factor, p53 mainly exerts its tumor-suppressive function through binding to consensus p53-responsive DNA elements (p53REs) in target genes to transcriptionally regulate these genes, which in turn regulates various crucial cellular processes[1–3]. To maintain its proper function, p53 is tightly regulated in cells mainly through post-translational modifications, especially the ubiquitination modification mediated by MDM2[1–3]. Therefore, p53 has a very short protein half-life and is maintained at a low level in cells under the non-stressed condition[1–3]. Loss of p53 function is a prerequisite for the initiation and/or progression of many cancers. The tumor-suppressive function of p53 is frequently disrupted by p53 mutations, which occur in over 50% of cancers. Although p53 mutations occur in all coding exons of the p53 gene, ~90% of p53 mutations are missense mutations that cluster in the DNA-binding domain (DBD)[3–6]. Furthermore, ~30% of missense mutations are localized at six "mutation hotspot" sites (e.g. 175, 245, 248, 249, 273, and 282) in the DBD[3–6]. Numerous studies have demonstrated that many missense mutant p53 (mutp53) proteins not only lose the function of wild-type p53 (wtp53) in tumor suppression due to the inability to bind to p53REs in target genes, but also gain new oncogenic activities to promote tumorigenesis independently of wtp53, termed as the gain-of-function (GOF) of mutp53[3–13]. While many different missense p53 mutations have been shown to display GOF activities in a context-dependent manner and it was estimated that the majority of missense p53 mutations may have different levels of GOF activities[14], these six hotspot mutations have been shown to display GOF activities and widely used as representatives for GOF studies[3–13]. Compelling evidence has shown that targeting mutp53 and its critical signaling

[1]Department of Radiation Oncology, Rutgers Cancer Institute, Rutgers-State University of New Jersey, New Brunswick, NJ, USA. [2]Department of Pathology, Princeton Medical Center, Princeton, NJ, USA. [3]Department of Cell Biology and Neuroscience, Rutgers-State University of New Jersey, Piscataway, NJ, USA. [4]Department of Biological Sciences, Hunter College, City University of New York, New York, NY, USA. [5]Center for Systems and Computational Biology, Rutgers Cancer Institute, Rutgers-State University of New Jersey, New Brunswick, NJ, USA. ✉e-mail: wh221@cinj.rutgers.edu; fengzh@cinj.rutgers.edu

pathways can effectively suppress cancer progression[3–13]. Intriguingly, missense mutp53 protein often becomes stabilized and accumulates to very high levels in cancers[3,5,6,8,12,13,15–17]. Positive immunohistochemistry (IHC) staining for p53 has been long and widely used as a surrogate for p53 mutational analysis in tumors in clinic[5,18–20]. Importantly, mutp53 accumulation has been demonstrated to be required for mutp53 GOF, and destabilizing mutp53 can greatly compromise mutp53 GOF and suppress cancer progression[3,5,6,8,12,13,15–17]. Given the unique feature of accumulation and GOF of mutp53 and the high mutational frequency of p53 in human cancers, targeting mutp53 for degradation in cancer cells has been actively tested as an attractive strategy that specifically targets cancer cells carrying mutp53 with minimal side effects on normal cells, which is urgently needed in cancer therapies[3,6,7]. Currently, the mechanisms of mutp53 accumulation and GOF in cancer are incompletely understood, and mutp53-targeted therapies are unavailable for cancer treatment[3,6,7]. Therefore, a better understanding of the mechanisms underlying mutp53 accumulation and GOF is crucial for the development of novel and effective strategies to treat cancers carrying mutp53.

Metabolic reprogramming is a hallmark of cancer cells[21–24]. Cancer cells reprogram metabolism to meet their bioenergetic and biosynthetic demands for rapid growth and adaptation to microenvironment stress, which is critical for cancer progression and a potential target for cancer therapies[21–24]. As a key metabolic change in many cancers, enhanced fatty acid synthesis provides cancer cells with phospholipids for cellular membranes, energy, and substrates for generating signaling molecules[21–23]. Lipogenic enzyme fatty acid synthase (FASN) catalyzes the conversion of acetyl-CoA and malonyl-CoA to palmitate (also known as palmitic acid), the 16-carbon long-chain saturated fatty acid, and plays a crucial role in fatty acid synthesis[23,24]. FASN is frequently overexpressed and/or activated in many cancers, including breast and colorectal cancers, which promotes fatty acid synthesis and cancer progression[21–25]. FASN is under active investigation as a potential target for cancer therapies[21–25]. In addition to its important role in fatty acid synthesis, FASN also plays a crucial role in S-palmitoylation, a post-translational modification of proteins in which palmitate, the direct product of FASN, is covalently linked to the thiol group of cysteine residues of proteins[26,27]. S-palmitoylation is mediated by a family of ZDHHC palmitoyltransferases containing a conserved Asp-His-His-Cys (DHHC) motif. A total of 23 human ZDHHCs have been identified[26,27]. S-palmitoylation modulates the stability, function and localization of proteins, which in turn regulates a variety of cellular processes under both physiological and pathological conditions, including development, immunity, inflammation, and cancer[26–30].

In this study, to reveal critical mechanisms underlying mutp53 protein accumulation in cancer, we screened for specific proteins interacting with mutp53 using liquid chromatography-tandem mass spectrometry (LC-MS/MS) assays, and identified FASN as an important mutp53-binding protein. Mutp53 has been suggested to be involved in the regulation of metabolism, including lipid metabolism[31]. However, its mechanism is not well understood. Given that FASN not only plays a critical role in lipid metabolism but also palmitoylation modification, which modulates the levels and functions of substrate proteins, we investigated the potential role of FASN in the regulation of mutp53 protein accumulation and GOF. Here, we show that FASN directly interacts with mutp53, but not wtp53, and enhances mutp53 palmitoylation, which in turn promotes mutp53 accumulation and GOF. Furthermore, blocking FASN suppresses mutp53 accumulation and GOF. Our results demonstrate that FASN-mediated mutp53 palmitoylation is an important mechanism for mutp53 accumulation and GOF, and suggest a potential therapeutic strategy for cancers carrying mutp53.

## Results

### FASN is a specific mutp53-interacting protein

*p53* missense mutations have two major categories: conformational mutants (e.g. R175H and G245S) and contact mutants (e.g. R248W and R273H)[3–5]. R175H, G245S, R248W and R273H hotspot mutp53 proteins display GOF activities and have been widely used for mutp53 GOF studies[3,6,8,12,13,15–17,32–34]. To reveal critical mechanisms underlying mutp53 accumulation and GOF, we screened for R175H mutp53-interacting proteins using co-immunoprecipitation (co-IP) followed by LC-MS/MS assays in human colorectal cancer *p53⁻/⁻* HCT116 cells expressing ectopic R175H mutp53 or wtp53. Through this strategy, FASN was identified as a potential mutp53-interacting protein (Supplementary Data 1). This interaction was confirmed by co-IP followed by Western blot assays in *p53⁻/⁻* HCT116 cells expressing ectopic FASN-Flag and R175H mutp53 or wtp53; FASN-Flag specifically interacted with R175H mutp53 but not wtp53 in cells (Fig. 1a). In addition, ectopic G245S, R248W and R273H mutp53 also interacted with FASN-Flag in *p53⁻/⁻* HCT116 cells although their interactions with FASN-Flag appeared to be weaker than R175H mutp53 (Fig. 1a).

The interactions between endogenous FASN and these hotspot mutp53 were observed by co-IP assays in different human cancer cell lines expressing homozygous endogenous mutp53, including breast cancer SK-BR3 (R175H) and MDA-MB468 (R273H), as well as colorectal cancer LS1034 (G245S) cells (Fig. 1b). The specific interaction between endogenous FASN and mutp53, but not wtp53, was confirmed in isogenic *p53^R248W/−^*, *p53^+/+^*, and *p53⁻/⁻* HCT116 cells, as well as *p53^R172H/R172H^* (equivalent to human R175H), *p53^+/+^* and *p53⁻/⁻* mouse embryonic fibroblasts (MEFs) (Fig. 1c). To determine the domain (s) of mutp53 required for the FASN-mutp53 interaction, serial deletion vectors expressing different domains of HA-tagged R175H mutp53, including transcriptional activation domains (AD1/2), DBD, tetramerization domain (TD), and C-terminal domain (CTD), were co-transfected with FASN-Flag vectors into *p53⁻/⁻* HCT116 cells for co-IP assays. Mutp53 DBD was shown to be required for the FASN-mutp53 interaction (Fig. 1d). The in vitro His pull-down assays using recombinant FASN-Flag and His-mutp53 or His-wtp53 proteins further showed that FASN directly interacted with mutp53, but not wtp53 (Fig. 1e). The interaction between endogenous FASN and mutp53, but not wtp53, in cells was further confirmed by proximity ligation assays (PLAs) detecting the in situ interaction of proteins in close proximity (<40 nm) in cells[35]. The FASN-mutp53 interaction signals were observed in SK-BR3 and *p53^R248W/−^* HCT116 cells but not SK-BR3 cells with mutp53 knockout (KO) by CRISPR/Cas9 technology or *p53^+/+^* and *p53⁻/⁻* HCT116 cells (Fig. 1f). Furthermore, the mutp53-FASN interaction signals were detected in both cytoplasmic and nuclear compartments with more interaction signals detected in the nuclear compartment (Fig. 1f). Our results from cytoplasmic/nuclear fractionation followed by Western blot analysis also showed the presence of FASN in the nuclear fraction, although more FASN protein was found in the cytoplasmic fraction in different cells (Supplementary Fig. 1). Similarly, FASN was reported to exhibit nuclear localization in prostate cancer cells[36]. These results together indicate that FASN specifically and directly interacts with GOF mutp53 but not wtp53 in cells.

### FASN promotes mutp53 protein accumulation in cells

FASN plays a critical role in protein palmitoylation through its production of palmitate, which can modulate protein stability[26]. We found that FASN directly interacts with mutp53, which is anticipated to result in an elevated local concentration of palmitate in close proximity to mutp53. Therefore, we tested whether FASN promotes mutp53 palmitoylation, leading to mutp53 accumulation. We first examined whether FASN promotes mutp53 accumulation in cells. FASN KO by CRISPR/Cas9 or FASN knockdown by shRNA vectors clearly decreased the mutp53 protein levels in different cells, including SK-BR3, MDA-MB468, and LS1034 (Fig. 2a, b). Furthermore, FASN

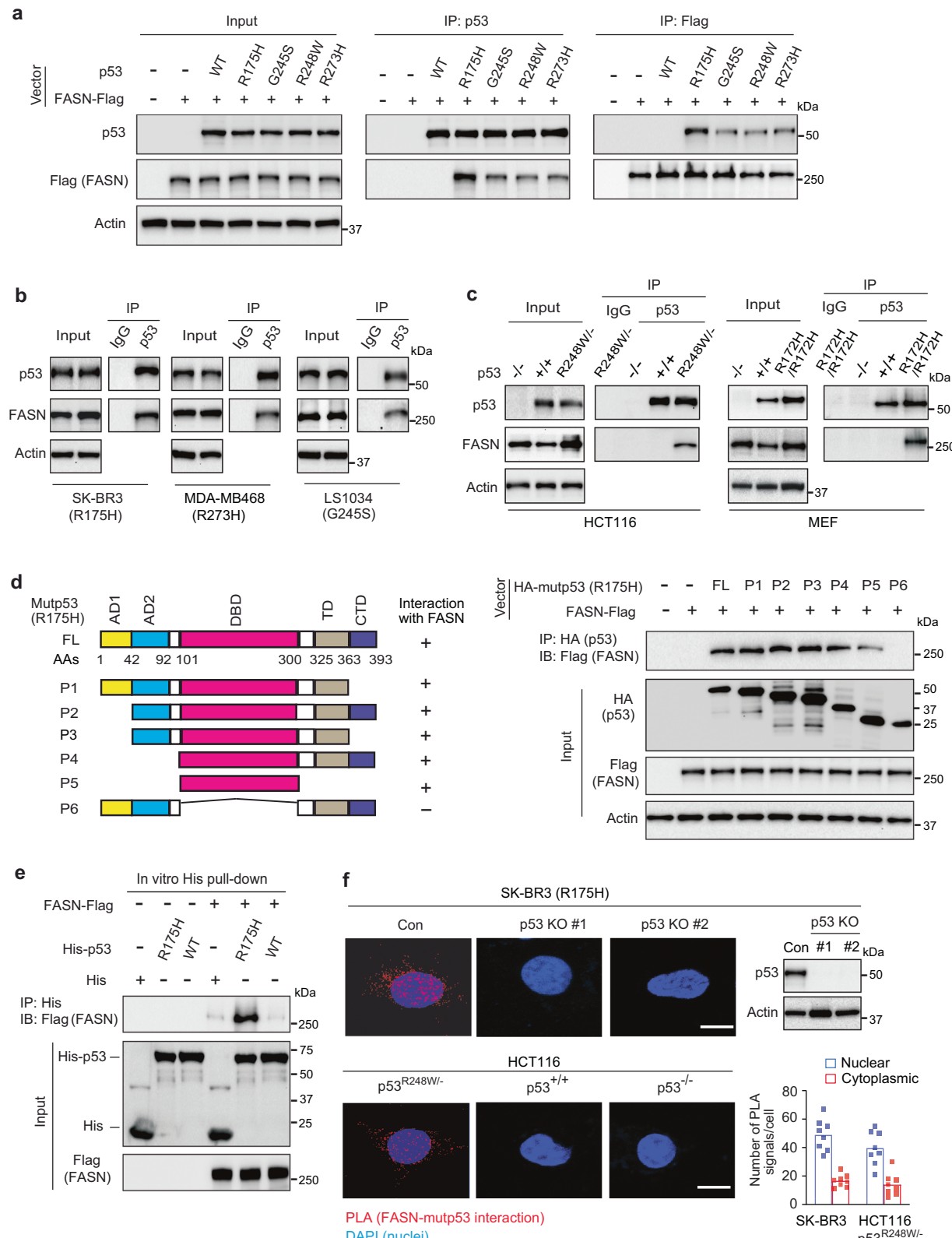

knockdown reduced mutp53 protein levels in $p53^{R248W/-}$ HCT116 cells and $p53^{R172H/R172H}$ MEFs, but not wtp53 protein levels in $p53^{+/+}$ HCT116 cells or $p53^{+/+}$ MEFs (Fig. 2c). Similarly, ectopic FASN expression promoted mutp53 accumulation in SK-BR3, MDA-MB468 and $p53^{R248W/-}$ HCT116 cells, but did not affect wtp53 protein levels in $p53^{+/+}$ HCT116 cells (Fig. 2d). Quantitative real-time PCR (qPCR) assays confirmed that

FASN did not affect mutp53 mRNA levels in cells (Supplementary Fig. 2a–c).

We further examined whether the promoting effect of FASN on mutp53 accumulation depends on its enzyme activity in fatty acid synthesis by using small-molecule FASN inhibitors. TVB-3166 is a specific small-molecule FASN inhibitor widely used in preclinical studies

**Fig. 1 | FASN is a specific mutp53-binding protein. a** Ectopic FASN protein specifically interacted with ectopic hotspot mutp53 but not wtp53 in cells. *p53⁻* HCT116 cells co-expressing FASN-Flag and ectopic wtp53 or different hotspot mutp53 were employed for co-IP assays. -: control vectors. WT: wild-type. **b** Co-IP analysis of the interaction of endogenous FASN with different endogenous hotspot mutp53 in SK-BR3, MDA-MB468, and LS1034 cells. **c** Endogenous FASN protein specifically interacted with endogenous hotspot mutp53 proteins, but not wtp53, in cells. Isogenic HCT116 cells and MEFs with different status of p53 were used for co-IP assays. **d** The DBD domain of R175H mutp53 is required for mutp53-FASN interaction. Left: schematic representation of vectors expressing full-length (FL) and serial deletion mutants of R175H HA-mutp53. AD: transactivation domain; DBD: DNA-binding domain; TD: tetramerization domain; CTD: C-terminal domain. Right:

*p53⁻* HCT116 cells co-expressing FASN-Flag and FL or deletion mutants of R175H HA-mutp53 were used for co-IP assays. **e** The direct interaction between mutp53 and FASN proteins analyzed by in vitro His pull-down assays using purified recombinant FASN-Flag and His-mutp53 (R175H) or His-wtp53 proteins. **f** The in situ interaction between endogenous mutp53 and FASN proteins in cells analyzed by the proximity ligation assay (PLA). Left panels: Represented PLA images. Red: PLA signals of the mutp53-FASN interaction in cells. Scale bar: 10 μm. Upper right panel: mutp53 KO by CRISPR/Cas9 in SK-BR3 cells. Lower right panel: the quantification of PLA signals in the nuclear and cytoplasmic compartments. Totally 200 cells were counted in 8 different areas. In **a–f**, all Western blot data represent three repeats with similar results. Source data are provided as a Source Data file.

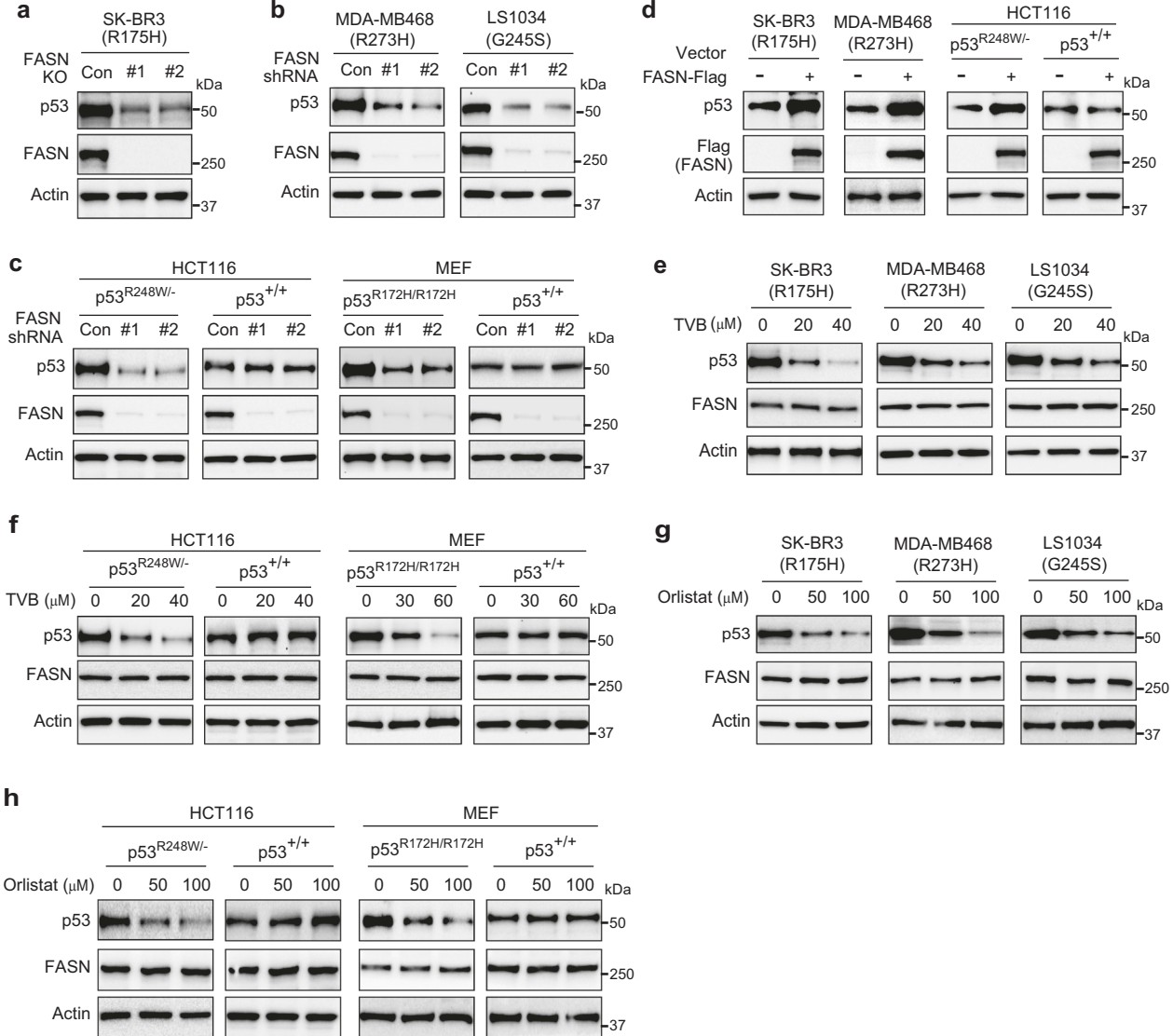

**Fig. 2 | FASN promotes mutp53 protein accumulation in cells. a** FASN KO reduced the levels of R175H mutp53 protein in SK-BR3 cells analyzed by Western blot assays. **b**, Knockdown of FASN by shRNA vectors reduced the protein levels of different hotspot mutp53 in MDA-MB468 and LS1034 cells. **c**, Knockdown of FASN by shRNA vectors reduced the mutp53 protein levels in *p53^R248W/-* HCT116 and *p53^R172H/R172H* MEF cells, but did not affect the wtp53 protein levels in *p53^+/+* HCT116 and *p53^+/+* MEF cells. Cells were transduced with control (Con) or 2 different lentiviral shRNA vectors against FASN for Western blot assays. **d** Ectopic FASN expression increased the protein levels of different hotspot mutp53, but not wtp53, in different cells. Cells were transduced with control (−) or FASN-Flag (+) expression

vectors for Western blot assays. **e**, **f** Inhibition of FASN by TVB-3166 reduced the protein levels of mutp53, but not wtp53, in different cells. Cells were treated with the small-molecule FASN inhibitor TVB-3166 (TVB) at the indicated concentrations for 24 h and subjected to Western blot analysis. **g**, **h** Inhibition of FASN by Orlistat reduced the protein levels of mutp53, but not wtp53, in different cells. Cells were treated with the small-molecule FASN inhibitor orlistat at the indicated concentrations for 24 h and then subjected to Western blot analysis. Data represent three repeats with similar results (**a–h**). Source data are provided as a Source Data file.

and the analog of TVB-2640, which is the first highly selective FASN inhibitor to enter clinical studies[37–40]. TVB-3166 treatment decreased mutp53 protein levels in different cells, including SK-BR3, MDA-MB468, LS1034 cells (Fig. 2e), $p53^{R248W/-}$ HCT116 cells, and $p53^{R172H/R172H}$ MEFs (Fig. 2f), but not wtp53 protein levels in $p53^{+/+}$ HCT116 cells or $p53^{+/+}$ MEFs (Fig. 2f). Similar effects were observed when cells were treated with Orlistat, another small-molecule FASN inhibitor[25,41]; Orlistat clearly reduced protein levels of mutp53, but not wtp53, in above-mentioned cells (Fig. 2g, h). These results suggest that FASN promotes mutp53 accumulation, which requires its enzyme activity in fatty acid synthesis.

## FASN promotes mutp53 palmitoylation

We further investigated whether FASN promotes mutp53 accumulation through enhancing mutp53 palmitoylation. First, we treated cells with 2-bromopalmitate (2-BP), a broad-spectrum ZDHHC inhibitor used as a palmitoylation inhibitor[42], to examine whether blocking palmitoylation reduces mutp53 levels. 2-BP treatment clearly decreased mutp53, but not wtp53, protein levels in cells (Fig. 3a). qPCR assays confirmed that 2-BP did not affect mutp53 mRNA levels in cells (Supplementary Fig. 3). Next, the acyl-biotin exchange (ABE) assay[43], a method widely used for the detection of S-palmitoylated proteins in cells, was employed to determine mutp53 palmitoylation in $p53^{-/-}$ HCT116 cells expressing different mutp53 or wtp53. The ABE assay involves blockade of unmodified cysteine thiol groups using N-ethylmaliemide (NEM), cleavage and unmasking of the thiol group of palmitoylated cysteines by hydroxylamine (HAM), and selective labeling of palmitoylated cysteines by biotin[43]. Palmitoylated p53 was detected in $p53^{-/-}$ HCT116 cells expressing different ectopic GOF mutp53 but not in control $p53^{-/-}$ HCT116 cells (Fig. 3b). Notably, mutp53 displayed much higher palmitoylation levels than wtp53 (Fig. 3b). Palmitoylation of endogenous mutp53 was also detected in different cells (Fig. 3c, Supplementary Fig. 4). Importantly, $p53^{R248W/-}$ HCT116 cells and $p53^{R172H/R172H}$ MEFs displayed much higher levels of palmitoylated p53 than their corresponding $p53^{+/+}$ cells (Fig. 3c). Furthermore, FASN knockdown by shRNAs or inhibition by TVB-3166 drastically decreased mutp53 palmitoylation and displayed a much stronger inhibitory effect on the palmitoylation of mutp53 than wtp53, suggesting that FASN plays a more important role in palmitoylation of mutp53 than wtp53 (Fig. 3d). Notably, TVB-3166 treatment did not affect the mutp53-FASN interaction (Supplementary Fig. 5).

To further investigate whether the impact of FASN inhibition by TVB-3166 on mutp53 palmitoylation is attributed to general palmitate depletion resulting from TVB-3166, we conducted a time-course study for TVB-3166 treatment to monitor mutp53 palmitoylation and ZDHHC3/17 autopalmitoylation. Notably, TVB-3166 treatment induced a much faster decrease in mutp53 palmitoylation than ZDHHC3/17 palmitoylation (Supplementary Fig. 6a). Interestingly, treating cells with exogenous palmitate, the direct product of FASN, moderately enhanced mutp53 palmitoylation and protein levels (Supplementary Fig. 6b, c), but did not fully rescue the decrease in mutp53 palmitoylation and protein levels resulting from FASN KO (Supplementary Fig. 6c-e). Notably, palmitate treatment did not affect the wtp53 protein levels in cells (Supplementary Fig. 6d). These results suggest that palmitate produced through FASN-mediated de novo lipogenesis is critical for mutp53 palmitoylation and protein levels.

The palmitoylation status of proteins can be modulated by the depalmitoylation process in addition to the palmitoylation process in cells[26,27]. To examine whether FASN affects the mutp53 depalmitoylation, we used bioorthogonal labeling with the palmitate analog 17-octadecynoic acid (17-ODYA) followed by coupling to a fluorescent azide-reporter tag[44]. FASN KO did not affect the rate of mutp53 depalmitoylation (Supplementary Fig. 7a). Depalmitoylating enzyme APT1 plays a critical role in protein depalmitoylation[26,27]. Palmostatin B,

a smallmolecule APT1 inhibitor[45], was employed to treat cells to block mutp53 depalmitoylation. Palmostatin B enhanced mutp53 palmitoylation, but did not rescue the decreased mutp53 palmitoylation induced by FASN KO or TVB-3166 treatment (Supplementary Fig. 7b, c). These results suggest that FASN promotes mutp53 palmitoylation mainly through regulating palmitoylation instead of depalmitoylation of mutp53.

S-palmitoylation modification occurs at cysteine residues of proteins[26,27]. p53 protein contains 10 cysteine residues, all of which are localized in the DBD (Fig. 3e). To identify the palmitoylation site(s) of mutp53, each of these 10 cysteine residues of R175H mutp53 was individually mutated to serine (CS mutations) and the palmitoylation levels of these individual CS mutants ectopically expressed in $p53^{-/-}$ HCT116 cells were analyzed by ABE assays. Compared with R175H mutp53, the C135S, C176S, C242S and C277S R175H mutp53 showed decreased palmitoylation levels, and furthermore, mutations of these 4 cysteines together (4CS mutation) almost completely abolished mutp53 palmitoylation, suggesting these 4 cysteines are the major palmitoylation sites of mutp53 (Fig. 3e, Supplementary Fig. 8a). Furthermore, 4CS mutations did not affect the mutp53-FASN interaction (Supplementary Fig. 8b).

## Palmitoylation inhibits mutp53 ubiquitination and degradation

The ubiquitination-mediated proteasomal degradation plays a critical role in downregulating mutp53 levels[5,15–17]. Therefore, we investigated whether FASN-mediated palmitoylation inhibits mutp53 ubiquitination and proteasomal degradation to promote mutp53 accumulation. SK-BR3 and $p53^{R248W/-}$ HCT116 cells were treated with 2-BP to block mutp53 palmitoylation, followed by treatment with the proteasome inhibitor MG132 to block ubiquitination-mediated proteasomal degradation, and then mutp53 protein levels were examined. While 2-BP clearly reduced mutp53 protein levels, MG132 increased mutp53 protein levels and largely abolished the inhibitory effect of 2-BP on mutp53 levels (Fig. 3f). Similar results were observed in cells with FASN KO or knockdown; MG132 increased mutp53 levels and largely abolished the inhibitory effect of FASN KO or knockdown on mutp53 levels (Fig. 3g). Notably, 4CS mutation reduced ectopic R175H mutp53 levels in $p53^{-/-}$ HCT116 cells, which was largely abolished by MG132 treatment (Fig. 3h). The ubiquitination assays showed that 2-BP treatment drastically increased mutp53 ubiquitination in SK-BR3 and $p53^{R248W/-}$ HCT116 cells (Fig. 3i), but not wtp53 in $p53^{+/+}$ HCT116 cells (Supplementary Fig. 9a). FASN KO or knockdown also drastically increased mutp53 ubiquitination in cells (Fig. 3j). Compared with R175H mutp53, 4CS R175H mutp53 showed increased ubiquitination (Fig. 3k). The effect of FASN-mediated palmitoylation on mutp53 protein stability was determined by measuring the mutp53 protein half-life using cycloheximide (CHX)-chase assays. FASN KO reduced mutp53 protein half-life in cells (Fig. 3l), and furthermore, 4CS R175H mutp53 showed a shorter protein half-life than R175H mutp53 (Fig. 3m). Collectively, these results suggest that FASN enhances mutp53 palmitoylation to block mutp53 ubiquitination and proteasomal degradation, promoting mutp53 stabilization and accumulation. Our results also suggest that FASN does not play a major role in wtp53 palmitoylation.

Palmitoylation regulates protein localization in addition to protein stability[26,27]. Interestingly, a recent study reported that wtp53 was palmitoylated by ZDHHC1, which did not affect wtp53 protein levels but enhanced the nuclear localization of wtp53 to promote the tumor-suppressive function of wtp53[46]. Immunofluorescence (IF) staining was used to examine whether FASN-mediated palmitoylation affects mutp53 localization in cells. Blocking palmitoylation by 2-BP reduced wtp53 nuclear localization as reported[46], but did not affect mutp53 nuclear localization in different cells, suggesting that palmitoylation has different biological consequences on wtp53 and mutp53 (Supplementary Fig. 9b–d).

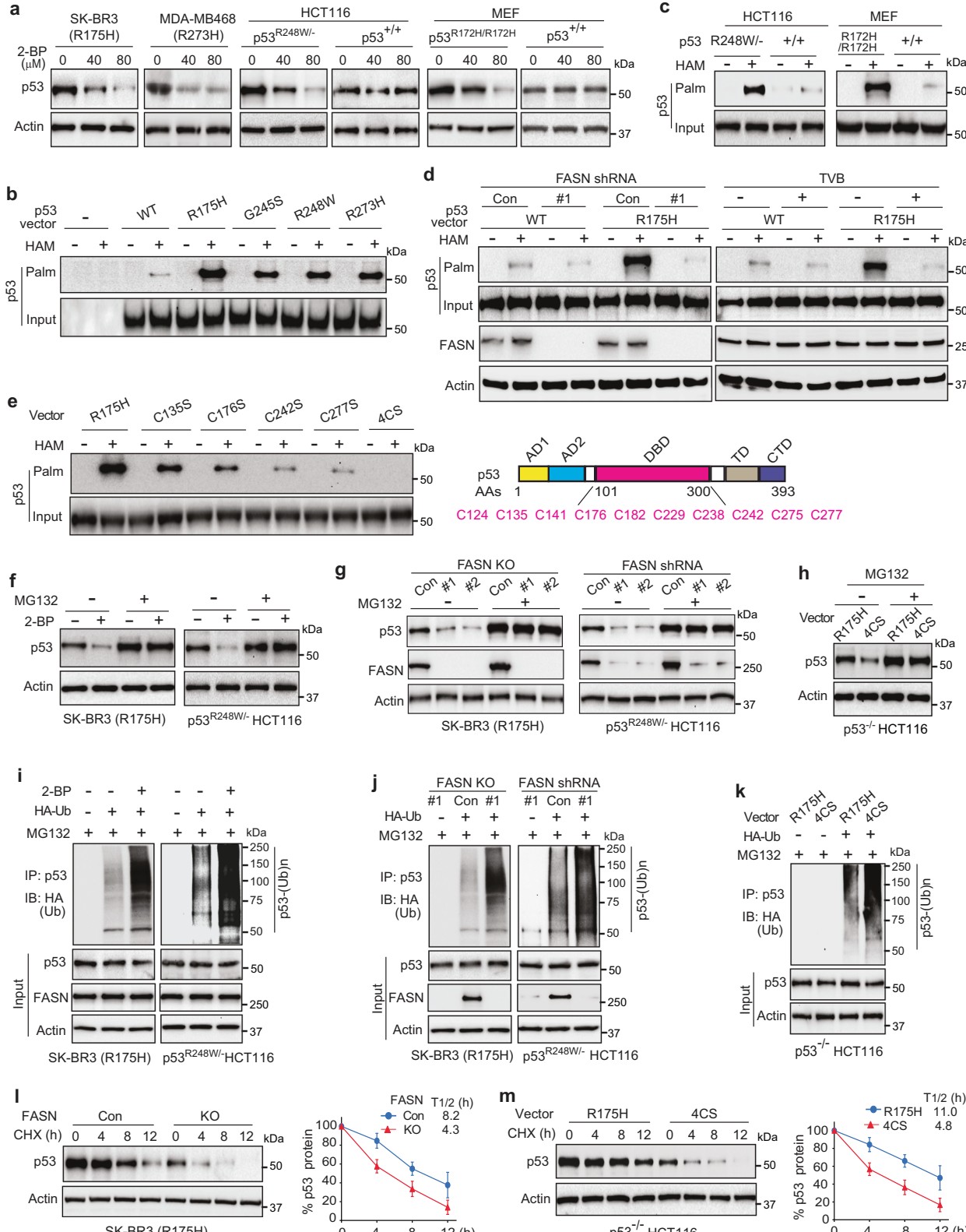

## Mutp53 is palmitoylated by ZDHHC3 and ZDHHC17

Palmitoylation is mediated by ZDHHC palmitoyltransferases, and 23 human ZDHHCs have been identified[26,27]. To screen for ZDHHC(s) that catalyzes mutp53 palmitoylation, we used small interfering RNAs (siRNAs) to knock down 23 ZDHHCs individually in SK-BR3 cells, and then measured mutp53 protein levels by Western blot assays. At least 2 different siRNAs with confirmed knockdown efficiency were used to target each ZDHHC[47]. Knockdown of ZDHHC3 and ZDHHC17, respectively, showed the most obvious inhibitory effect on mutp53 protein levels in SK-BR3 cells (Supplementary Fig. 10a, b and Fig. 4a, b), which was also observed in MDA-MB468 cells (Fig. 4a, b). Further, ectopic expression of ZDHHC3 and/or ZDHHC17 increased mutp53 protein levels in SK-BR3, MDA-MB468 and *p53*[R248W/-] HCT116 cells, but did not affect wtp53 protein levels in *p53*[+/+] HCT116 cells (Fig. 4c). ABE assays

**Fig. 3 | FASN enhances mutp53 palmitoylation to suppress mutp53 ubiquitination and stabilize mutp53. a** The palmitoylation inhibitor 2-BP reduced protein levels of mutp53, but not wtp53, in different cells. Cells were treated with the 2-BP for 24 h before Western blot analysis. ABE analysis of palmitoylation of ectopic (**b**) or endogenous (**c**) mutp53 and wtp53 in cells. In **b**, $p53^{-/-}$ HCT116 cells expressing ectopic wtp53 or different mutp53 were employed for ABE assays. -: control vectors. HAM: hydroxylamine. Palm p53: palmitoylated p53. **d** FASN knockdown (left) or inhibition by TVB-3166 (TVB; right) drastically reduced mutp53 palmitoylation in $p53^{-/-}$ HCT116 cells determined by ABE assays. **e**, Identification of mutp53 palmitoylation sites by ABE assays in $p53^{-/-}$ HCT116 cells expressing R175H mutp53 or its different CS (cysteine to serine) mutants. Right panel: cysteine residues of p53. In **b-e**, to enable the standardized comparison of the p53 palmitoylation levels across various samples, the input p53 levels were normalized by adding different amounts of protein samples for Western blot analysis. In **d**, Actin was used as the control for FASN only. MG132 treatment largely abolished the inhibitory effect of 2-BP (**f**), FASN KO or knockdown (**g**), and the 4CS mutation (**h**) on mutp53 protein levels in cells. 2-BP treatment (**i**), FASN KO or knockdown (**j**), and the 4CS mutation (**k**) promoted mutp53 ubiquitination in cells analyzed by ubiquitination assays. Ub: Ubiquitin. FASN KO (**l**) and the 4CS mutation (**m**) reduced mutp53 protein half-life ($T_{1/2}$) in cells. Cells were treated with CHX for different hours before Western blot assays. Data represent mean ± SD (n = 3 independent experiments). In **a-m**, Western blot data represent three repeats with similar results. Source data are provided as a Source Data file.

showed that knockdown of ZDHHC3 and/or ZDHHC17 greatly reduced mutp53 palmitoylation (Fig. 4d). In contrast, ectopic expression of ZDHHC3 and/or ZDHHC17 in cells enhanced mutp53 palmitoylation (Fig. 4e). Co-IP assays showed that ZDHHC3 and ZDHHC17 specifically interacted with mutp53 but not wtp53 in cells (Fig. 4f, g). The in vitro GST pull-down assays using recombinant GST-ZDHHC3/17 and His-mutp53 or His-wtp53 proteins further showed that ZDHHC3 and ZDHHC17 directly interacted with mutp53 but not wtp53 (Fig. 4h). Notably, knockdown of ZDHHC3 and ZDHHC17 together by siRNAs did not affect the mutp53-FASN interaction (Supplementary Fig. 11a), and 4CS mutations did not affect the interaction between mutp53 and ZDHHC3/17 (Supplementary Fig. 11b, c). The cytoplasmic/nuclear fractionation analysis showed the presence of ZDHHC3 and ZDHHC17 in both cytoplasmic and nuclear fractions (Supplementary Fig. 12a). PLAs confirmed that mutp53 and ZDHHC3/17 interactions were detected in both cytoplasmic and nuclear compartments with more interactions in the nuclear compartment (Supplementary Fig. 12b). These results together indicate that ZDHHC3 and ZDHHC17 play major roles in catalyzing mutp53 palmitoylation.

## FASN promotes mutp53 GOF in tumorigenesis

Mutp53 accumulation is crucial for its GOF activities in tumorigenesis[3,6,13,15-17]. Soft agar colony formation assays detecting the ability of anchorage-independent growth (AIG) of cancer cells is a well-established assay for cellular transformation/tumorigenicity. We examined the effect of FASN on mutp53 GOF using AIG assays in different cell lines. Consistent with previous reports[48-51], GOF mutp53 significantly promoted AIG, whereas wtp53 inhibited AIG of cancer cells (Fig. 5a-d). Similarly, FASN knockdown exerted a more pronounced inhibitory effect on AIG in $p53^{R248W/-}$ HCT116 cells and E1A/Ras-transformed $p53^{R172H/R172H}$ MEFs compared with their corresponding $p53^{+/+}$ or $p53^{-/-}$ cells (Fig. 5c). In addition, ectopic FASN expression displayed a more pronounced promoting effect on AIG in SK-BR3 and $p53^{R248W/-}$ HCT116 cells compared with SK-BR3 cells with mutp53 KO and $p53^{+/+}$ or $p53^{-/-}$ HCT116 cells, respectively (Fig. 5d). Notably, while FASN KO or knockdown significantly reduced AIG of SK-BR3, MDA-MB468 and LS1034 cells, mutp53 KO or knockdown greatly abolished the inhibitory effect of FASN KO or knockdown on AIG (Fig. 5a, b and Supplementary Fig. 13). SK-BR3 cells with FASN KO were stably transduced with FASN expression vectors to increase the level of FASN protein to a comparable level of endogenous FASN in control SK-BR3 cells and used for AIG assays. Restoration of FASN expression in SK-BR3 cells with FASN KO rescued the decreased mutp53 protein levels and impaired AIG in soft agar of cells resulting from FASN KO, and furthermore, displayed a more pronounced promoting effect on AIG in SK-BR3 cells compared with SK-BR3 cells with mutp53 KO (Supplementary Fig. 14a, b). These results together suggest that promoting mutp53 accumulation and GOF through palmitoylation is an important mechanism for FASN to enhance AIG in addition to the known lipogenic mechanism of FASN.

GOF mutp53 promotes tumorigenesis in xenograft tumor models[5,8,48-52]. Here, we examined the effect of FASN on mutp53 GOF in

tumorigenesis using orthotopic breast tumors formed by the fat pad injection of SK-BR3 cells in athymic nude mice. Mutp53 KO in SK-BR3 cells significantly inhibited the growth of breast orthotopic tumors, demonstrating the GOF of R175H mutp53 in tumorigenesis (Fig. 5e). FASN KO in cells significantly inhibited the growth of breast orthotopic tumors formed by SK-BR3 cells, and notably, displayed a much more pronounced inhibitory effect on tumors formed by SK-BR3 cells compared with tumors formed by SK-BR3 cells with mutp53 KO (Fig. 5e). Western blot and IHC staining analysis of tumor tissues showed that FASN KO decreased mutp53 protein levels in tumors (Fig. 5f, g). Notably, ectopic expression of FASN in cells with FASN KO rescued the impaired growth of breast orthotopic tumors resulting from FASN KO (Supplementary Fig. 14c). Similar results were observed in subcutaneous (s.c.) tumors formed by isogenic HCT116 cells. Compared with $p53^{-/-}$ HCT116 tumors, $p53^{R248W/-}$ HCT116 tumors displayed a growth advantage whereas $p53^{+/+}$ HCT116 tumors displayed a growth disadvantage (Fig. 5h). FASN knockdown in cells displayed a more pronounced inhibitory effect on $p53^{R248W/-}$ HCT116 tumors compared with $p53^{+/+}$ or $p53^{-/-}$ HCT116 tumors (Fig. 5h). Furthermore, FASN knockdown reduced mutp53 protein levels in $p53^{R248W/-}$ HCT116 tumors but not wtp53 protein levels in $p53^{+/+}$ HCT116 tumors (Fig. 5i). These results demonstrate that FASN enhances mutp53 accumulation and GOF activities in promoting AIG in vitro and tumor growth in vivo.

It has been well-established that FASN overexpression and/or activation plays an important role in promoting tumorigenesis through lipogenesis to provide fatty acids for membrane synthesis and energy for cancer cells[23-25]. We investigated whether restoration of mutp53 expression in cells with FASN KO rescues the impaired AIG and tumorigenesis resulting from FASN KO. SK-BR3 cells with FASN KO were stably transduced with R175H mutp53 vectors to increase the level of mutp53 protein to a comparable level of endogenous mutp53 in control SK-BR3 cells without FASN KO, and then employed for AIG and tumorigenesis assays. Restoration of R175H mutp53 expression in cells with FASN KO significantly but did not fully rescue the impaired AIG and tumorigenesis resulting from FASN KO (Supplementary Fig. 15a-c), suggesting that while the palmitoylation of mutp53 by FASN contributes greatly to the promoting effect of FASN on AIG and tumorigenesis, the known functions of FASN, especially in lipogenesis, also play important roles in promoting AIG and tumorigenesis.

## Pharmacological inhibition of FASN preferentially suppresses tumorigenesis of mutp53 cancer cells

To confirm the role of FASN in regulating mutp53 GOF in cancer cells and test the possibility of targeting FASN as a potential strategy to treat cancers carrying mutp53, we evaluated the impact of pharmacological inhibition of FASN by TVB-3166 on AIG. TVB-3166 treatment exerted a much more pronounced inhibitory effect on AIG of SK-BR3, MDA-MB468, and LS1034 cells compared with their corresponding cells with mutp53 KO or knockdown (Fig. 6a, b). Furthermore, TVB-3166 displayed a more pronounced inhibitory effect on AIG of $p53^{R248W/-}$ HCT116 cells and E1A/Ras-transformed $p53^{R172H/R172H}$ MEFs compared with their isogenic $p53^{+/+}$ or $p53^{-/-}$ cells (Fig. 6c). Similar results were observed

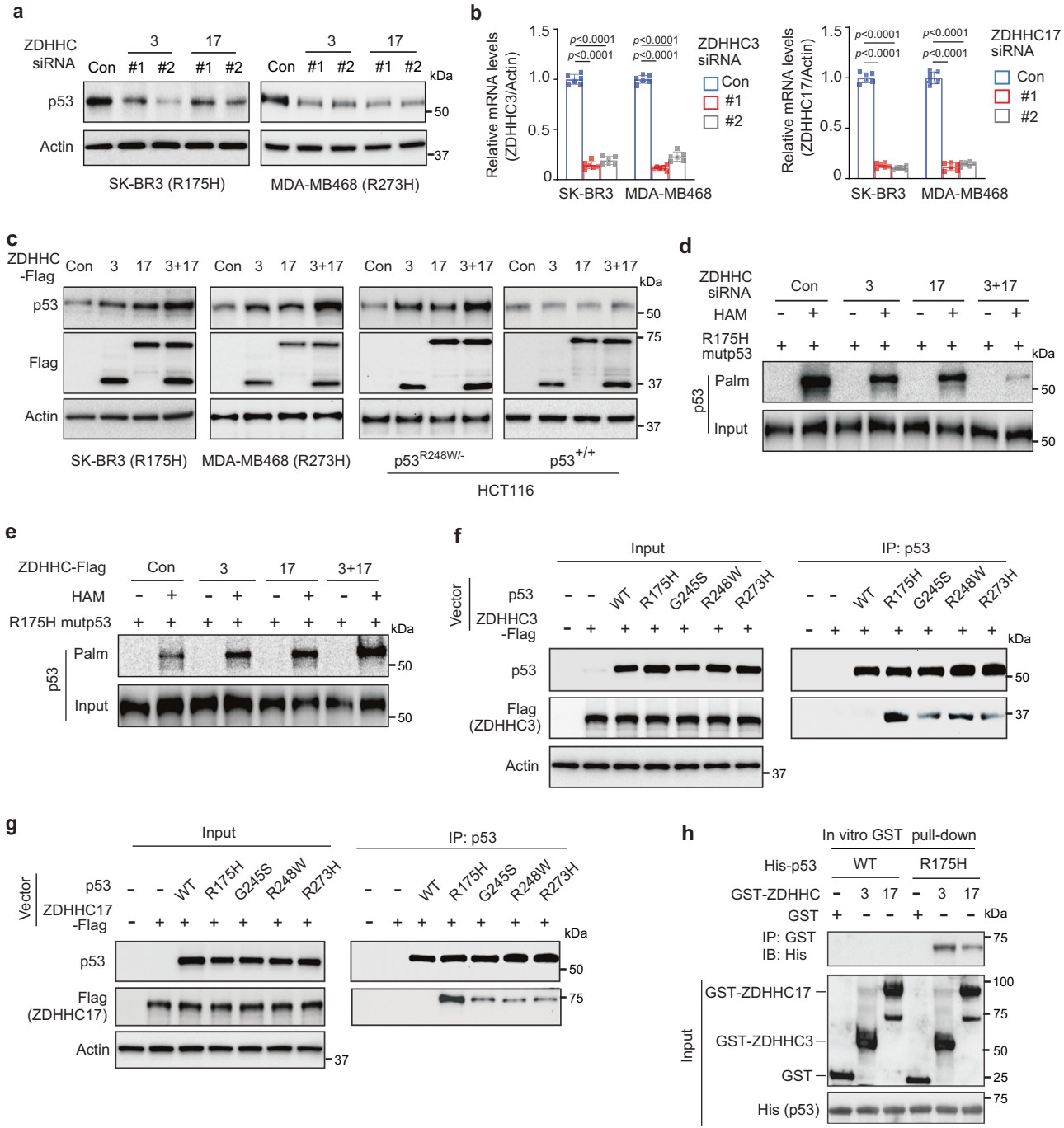

**Fig. 4 | Mutp53 is palmitoylated by ZDHHC3 and ZDHHC17 palmitoyl-transferases.** Knockdown of ZDHHC3 or ZDHHC17 by siRNAs reduced mutp53 protein levels in cells. Mutp53 protein levels were analyzed by Western blot assays (**a**), and the knockdown of ZDHHC3 and ZDHHC17 by siRNAs was analyzed by qPCR assays (**b**). Data represent mean ± SD ($n = 6$ independent experiments), one-way ANOVA followed by Dunnett's test. **c** Ectopic expression of ZDHHC3 and/or ZDHHC17 increased the protein levels of mutp53, but not wtp53, in cells. **d** Knockdown of ZDHHC3 and/or ZDHHC17 reduced mutp53 palmitoylation in $p53^{-/-}$ HCT116 cells expressing ectopic R175H mutp53 analyzed by ABE assays. **e** Ectopic expression of ZDHHC3 and/or ZDHHC17 enhanced mutp53 palmitoylation in $p53^{-/-}$ HCT116 cells expressing ectopic R175H mutp53 together with ZDHHC3 and/or

ZDHHC17 analyzed by ABE assays. In **d, e**, to enable the standardized comparison of the p53 palmitoylation levels across various samples, the input p53 protein levels were normalized by adding different amounts of protein samples for Western blot analysis. ZDHHC3 (**f**) and ZDHHC17 (**g**) proteins specifically interacted with different hotspot mutp53, but not wtp53, in cells analyzed by co-IP assays. $p53^{-/-}$ HCT116 cells expressing ZDHHC3 (**f**) or ZDHHC17 (**g**) together with wtp53 or different hotspot mutp53 were used for co-IP assays. -: control vectors. **h** The direct interaction between mutp53 and ZDHHC3 or ZDHHC17 proteins analyzed by in vitro GST pull-down assays using recombinant His-mutp53 (R175H) or His-wtp53 and GST-ZDHHC3 or GST-ZDHHC17 proteins. Data represent three repeats with similar results (**a, c–h**). Source data are provided as a Source Data file.

when cells were treated with FASN inhibitor Orlistat (Supplementary Fig. 16).

We further examined the effect of TVB-3166 on mutp53 GOF in promoting tumor growth. Mice bearing orthotopic breast tumors were

treated with TVB-3166 (60 mg/kg, oral gavage, once every two days for 3 weeks) when tumor volumes reached ~30 mm³. TVB-3166 exerted a more pronounced inhibitory effect on the growth of orthotopic tumors formed by SK-BR3 cells than tumors formed by SK-BR3 cells

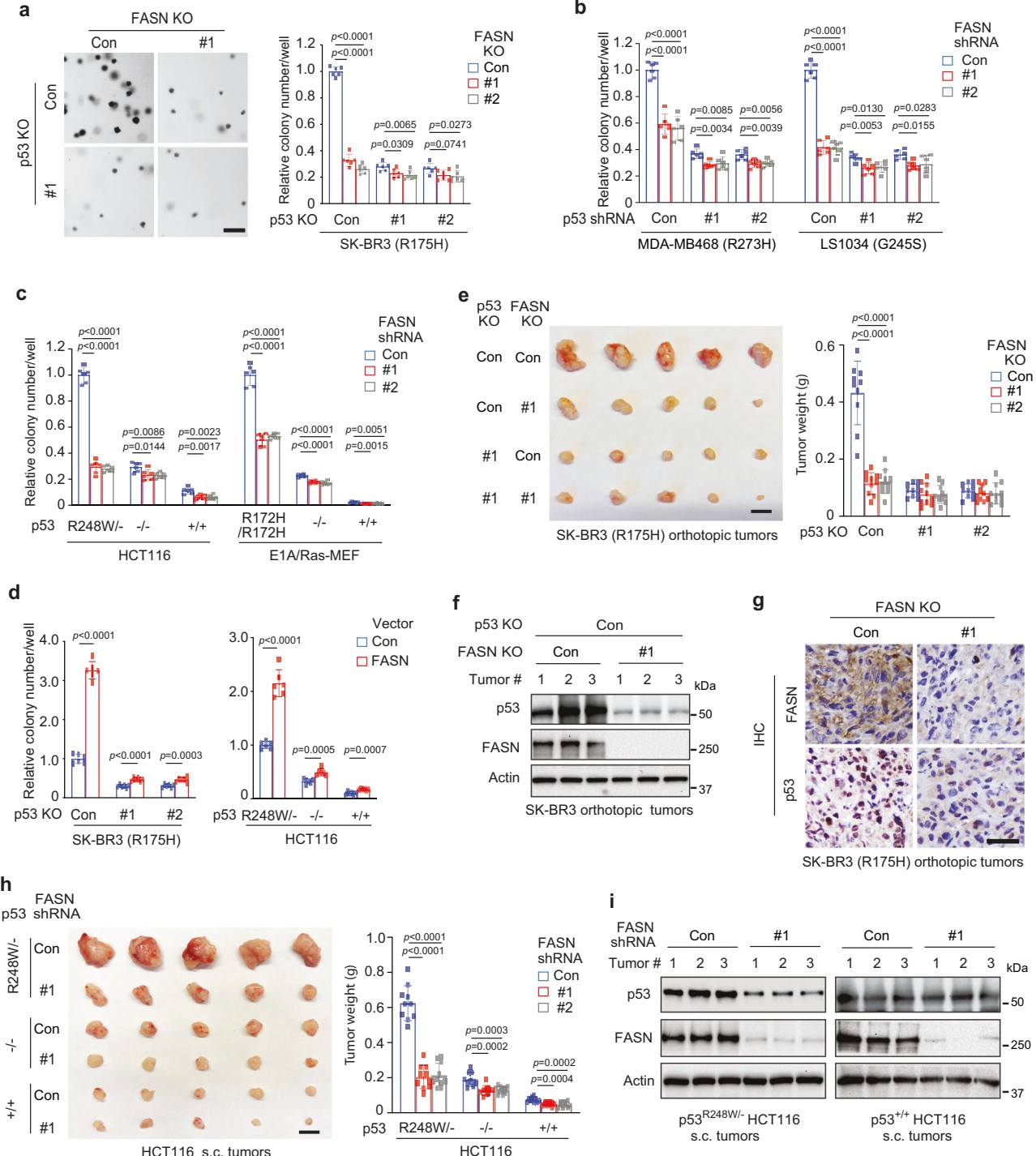

**Fig. 5 | FASN enhances mutp53 GOF in promoting anchorage-independent growth (AIG) and tumorigenesis. a** FASN KO displayed a more pronounced inhibitory effect on AIG in SK-BR3 cells compared with SK-BR3 with mutp53 KO. Left: Representative images of AIG of cells. Scale bar: 200 μm. **b** FASN knockdown displayed a more pronounced inhibitory effect on AIG in MDA-MB468 and LS1034 cells compared with corresponding cells with mutp53 knockdown. **c** FASN knockdown displayed a more pronounced inhibitory effect on AIG in *p53^R248W/-* HCT116 and E1A/Ras-transformed *p53^R172H/R172H* MEF cells compared with their corresponding *p53^+/+* or *p53^-/-* cells. **d** Ectopic FASN expression displayed a more pronounced promoting effect on AIG of SK-BR3 and *p53^R248W/-* HCT116 cells compared with SK-BR3 cells with mutp53 KO and *p53^+/+* or *p53^-/-* HCT116 cells, respectively. **e** FASN KO displayed a more pronounced inhibitory effect on the growth of orthotopic tumors formed by SK-BR3 cells compared with tumors formed by SK-BR3 cells with mutp53

KO. Left: The image of collected tumors. Scale bar: 10 mm. Right: The weights of collected tumors. FASN KO reduced mutp53 protein levels in SK-BR3 tumors analyzed by Western blot (**f**) and IHC staining (**g**) analysis. Scale bar: 40 μm. **h** FASN knockdown displayed a more pronounced inhibitory effect on the growth of tumors formed by *p53^R248W/-* HCT116 cells compared with tumors formed by *p53^+/+* or *p53^-/-* HCT116 cells. Left: The image of collected tumors. Scale bar: 10 mm. **i** FASN knockdown reduced mutp53 protein levels in *p53^R248W/-* HCT116 tumors but not wtp53 protein levels in *p53^+/+* HCT116 tumors. In **f**, **i**, similar results were observed in samples from at least 6 mice/group, and results from 3 mice/group are presented. Data represent mean ± SD (n = 6 independent experiments in **a–d**; n = 10 mice/group in **e**, **h**). In d, two-tailed unpaired Student's *t*-test. In a, b, c, e, and h, one-way ANOVA followed by Dunnett's test. Source data are provided as a Source Data file.

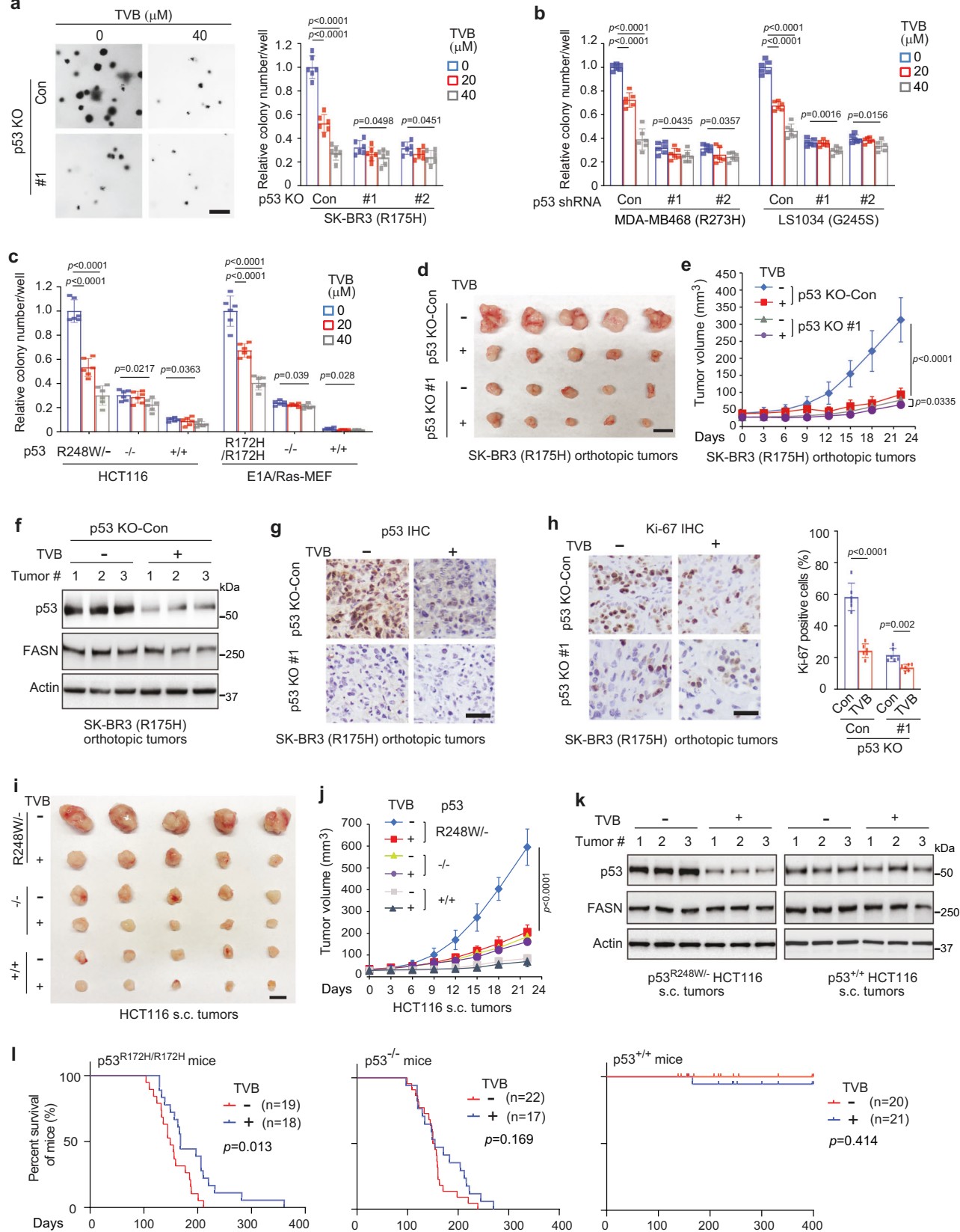

with mutp53 KO (Fig. 6d, e). TVB-3166 reduced mutp53 protein levels and inhibited tumor cell proliferation as reflected by the percentage of Ki-67-positive cells in tumors (Fig. 6f-h). Similarly, TVB-3166 displayed a more pronounced inhibitory effect on the growth of *p53^R248W/−* HCT116 s.c. tumors than *p53^+/+* or *p53^−/−* HCT116 s.c. tumors (Fig. 6i, j),

and reduced mutp53 protein levels in *p53^R248W/−* HCT116 tumors but not wtp53 protein levels in *p53^+/+* HCT116 tumors (Fig. 6k). Notably, TVB-3166 did not clearly affect the body weight of mice (Supplementary Fig. 17). Regulating gene expression is an important mechanism for mutp53 to exert its GOF[3,5,53]. The qPCR assays showed that KO of R175H

**Fig. 6 | Pharmacological inhibition of FASN preferentially suppresses AIG of cancer cells carrying mutp53 and growth of tumors carrying mutp53.** The FASN inhibitor TVB-3166 displayed a more pronounced inhibitory effect on AIG in cancer cells carrying mutp53 compared with cells with mutp53 KO (**a**) or knockdown (**b**). Left in **a**: Representative images. Scale bar: 200 μm. **c** TVB-3166 displayed a more pronounced inhibitory effect on AIG in $p53^{R248W/-}$ HCT116 and E1A/Ras-transformed $p53^{R172H/R172H}$ MEFs compared with their corresponding $p53^{+/+}$ or $p53^{-/-}$ cells. In **a–c**, cells were treated with or without TVB-3166 for 72 h. **d**, **e** TVB-3166 displayed a more pronounced inhibitory effect on the growth of orthotopic tumors formed by SK-BR-3 (R175H) cells than tumors formed by SK-BR3 with mutp53 KO. Mice were treated with TVB-3166 once every two days for 3 weeks when tumor sizes reached -30 mm³. **d** Scale bar: 10 mm. **e** the growth curve of tumors. TVB-3166 reduced mutp53 protein levels (**f**, **g**) and Ki-67 positive staining (**h**) in SK-BR3 tumors analyzed by Western blot analysis (**f**) and IHC staining (**g**, **h**), respectively. Scale bar: 40 μm. **i**, **j** TVB-3166 displayed a more pronounced inhibitory effect on the growth of $p53^{R248W/-}$ HCT116 s.c. tumors compared with $p53^{+/+}$ or p53^{-/-} s.c. tumors. **k** TVB-3166 reduced mutp53 protein levels in p53^{R248W/-} HCT116 tumors, but not wtp53 protein levels in $p53^{+/+}$ tumors. In **f**, **k**, similar results were observed in tumors from at least 6 mice/group, and results from 3 mice/group are presented. Data represent mean ± SD. In **a–c**, $n = 6$ independent experiments, one-way ANOVA followed by Dunnett's test. In **e**, **j**, $n = 10$ mice/group, two-way ANOVA followed by Bonferroni's tests. In **h**, $n = 6$ independent experiments, two-tailed unpaired Student's $t$-test. **l** TVB-3166 extended the lifespan in $p53^{R172H/R172H}$ mice, but not in p53^{-/-} mice. Presented are Kaplan-Meier survival curves of eight-week-old mice treated with vehicle (−) or TVB-3166 once every 2 days for 12 weeks. Analyzed by the log-rank (Mantel-Cox) test. TVB: TVB-3166. Source data are provided as a Source Data file.

mutp53 significantly reduced the expression of several well-known mutp53-regulated genes, including ADRB1, IGFBP3 and P2RX5[3,5,53], in SK-BR3 tumors (Supplementary Fig. 18). Notably, TVB-3166 significantly reduced the expression of these genes in SK-BR3 tumors carrying mutp53 but not in tumors with mutp53 KO (Supplementary Fig. 18). These results indicate that pharmacological inhibition of FASN suppresses mutp53 accumulation and GOF and preferentially inhibits the growth of mutp53 tumors compared with $p53^{+/+}$ or $p53^{-/-}$ tumors.

Consistent with our observation that TVB-3166 treatment reduced mutp53 protein levels in SK-BR3 tumors (Fig. 6f), TVB-3166 also drastically reduced mutp53 palmitoylation in tumors (Supplementary Fig. 19a; collected one day after the completion of TVB-3166 treatment). However, the decreases in mutp53 palmitoylation and protein levels largely disappeared in the tumors collected ten days after the completion of TVB-3166 treatment compared with tumors without TVB-3166 treatment (Supplementary Fig. 19b, c), indicating that the effect of FASN inhibition on mutp53 palmitoylation and protein levels in tumors gradually disappears after TVB-3166 withdrawal.

### Pharmacological inhibition of FASN preferentially suppresses tumorigenesis in $p53^{R172H/R172H}$ mice

$p53^{R172H/R172H}$ and $p53^{-/-}$ mice develop spontaneous tumors (mainly lymphomas and sarcomas) at early ages, and therefore, have much shorter lifespans compared with $p53^{+/+}$ mice[15,54,55]. Intriguingly, while mutp53 specifically accumulates in tumors, but not in normal tissues, in $p53^{R172H/R172H}$ mice, further deletion of E3 ubiquitin ligases for mutp53, such as MDM2 or TRIM21, in p53^{R172H/R172H} mice results in mutp53 accumulation in normal tissues and tumors, an earlier tumor onset, and the reduced lifespan of mice, indicating that mutp53 accumulation is a tumor-specific event and mutp53 accumulation is required for mutp53 GOF in tumorigenesis[15,17,54,55]. To further evaluate the effect of pharmacological inhibition of FASN on mutp53 GOF in tumorigenesis, 8-week-old $p53^{R172H/R172H}$, $p53^{+/+}$, and $p53^{-/-}$ mice were treated with TVB-3166 (60 mg/kg, oral gavage, once every two days) for 12 weeks. While TVB-3166 did not affect the tumor spectrum in $p53^{R172H/R172H}$ mice (mainly lymphomas and sarcomas; Supplementary Table 1), TVB-3166 suppressed tumorigenesis and significantly extended the lifespan in $p53^{R172H/R172H}$ mice but not $p53^{-/-}$ mice (Fig. 6l). Consistent with our observation that the effect of TVB-3166 on mutp53 palmitoylation and protein levels in orthotopic tumors gradually disappeared at 10 days after the completion of TVB-3166 (Supplementary Fig. 19b, c), there was no clear difference in mutp53 palmitoylation levels between the tumors collected from $p53^{R172H/R172H}$ mice treated with or without TVB-3166 (Supplementary Fig. 19d). It is worth mentioning that majority of $p53^{R172H/R172H}$ mice succumbed to mortality far more than ten days after the completion of TVB-3166 treatment (12 weeks) when their tumors were collected for analysis (Fig. 6l). Collectively, these results suggest that FASN inhibition by TVB-3166 suppresses mutp53 GOF and preferentially inhibits growth of mutp53 tumors to extend the lifespan of $p53^{R172H/R172H}$ mice.

### Pharmacological inhibition of FASN preferentially suppresses growth of mutp53 human colorectal cancer organoids

Patient-derived organoids mimic the biological characteristics of the primary tumors, and have been used to predict the most suitable treatment for cancer patients. To confirm our results that FASN inhibition preferentially suppresses the growth of mutp53 tumors in xenograft tumor models and mutp53 knock-in mice, different human colorectal cancer organoids expressing mutp53 or wtp53 were used to evaluate the effect of TVB-3166. TVB-3166 treatment significantly decreased the viability of organoids expressing R175H, R248Q, and R282W mutp53, respectively, and furthermore, displayed a more pronounced inhibitory effect on these organoids than organoids expressing wtp53 (Fig. 7a). Western blot analysis showed that TVB-3166 reduced mutp53 but not wtp53 protein levels in organoids (Fig. 7b). Furthermore, mutp53 KO by CRISPR/Cas9 in mutp53 organoids greatly reduced the inhibitory effect of TVB-3166 on these organoids (Fig. 7c). These results indicate that human colorectal cancer organoids carrying mutp53 are more sensitive to TVB-3166 than $p53^{+/+}$ or $p53^{-/-}$ organoids.

### The association of FASN and mutp53 expression in human cancer tissues

The wtp53 protein has a very short half-life and is maintained at a low level in cells under the non-stressed condition mainly through the ubiquitination modification mediated by MDM2 and other E3 ubiquitin ligases[1–3]. Due to the feature of mutp53 accumulation in tumors, positive p53 IHC staining has been long used as a surrogate for p53 mutational analysis in tumors in the clinic, although it is not a perfect marker[5,18–20]. Similar results were also observed in our previous studies to analyze p53 mutations in human colorectal tumor specimens using both p53 IHC staining and direct sequencing of the p53 gene[16] (Supplementary Fig. 20). Around 70-80% of triple negative breast cancer (TNBC) and ~40-60% of colorectal cancer carry p53 mutations[3–5]. Here, we further investigated whether FASN increases mutp53 levels in human cancer specimens by analyzing the association between FASN expression levels and mutp53 levels in TNBC and colorectal tumor tissues. FASN and p53 protein levels in tumor tissues were determined by IHC staining using tumor tissue arrays (TMAs). Positive p53 staining was observed in 78.3% of TNBC (144 out of 184) and 53.7% of colorectal tumors (158 out of 294), respectively, and a very significant correlation between the high levels of FASN and p53 were found in TNBC and colorectal tumor tissues (Fig. 7d, e), suggesting that FASN overexpression is an important mechanism for mutp53 protein accumulation in cancers.

## Discussion

p53 is mutated in over 50% of cancers and almost every type of cancer. Mutp53 often accumulates to high levels in cancer cells to promote cancer progression through GOF mechanisms. Therefore, mutp53 is an attractive target for cancer therapies[3,5,7,8,13]. However, so far, mutp53-

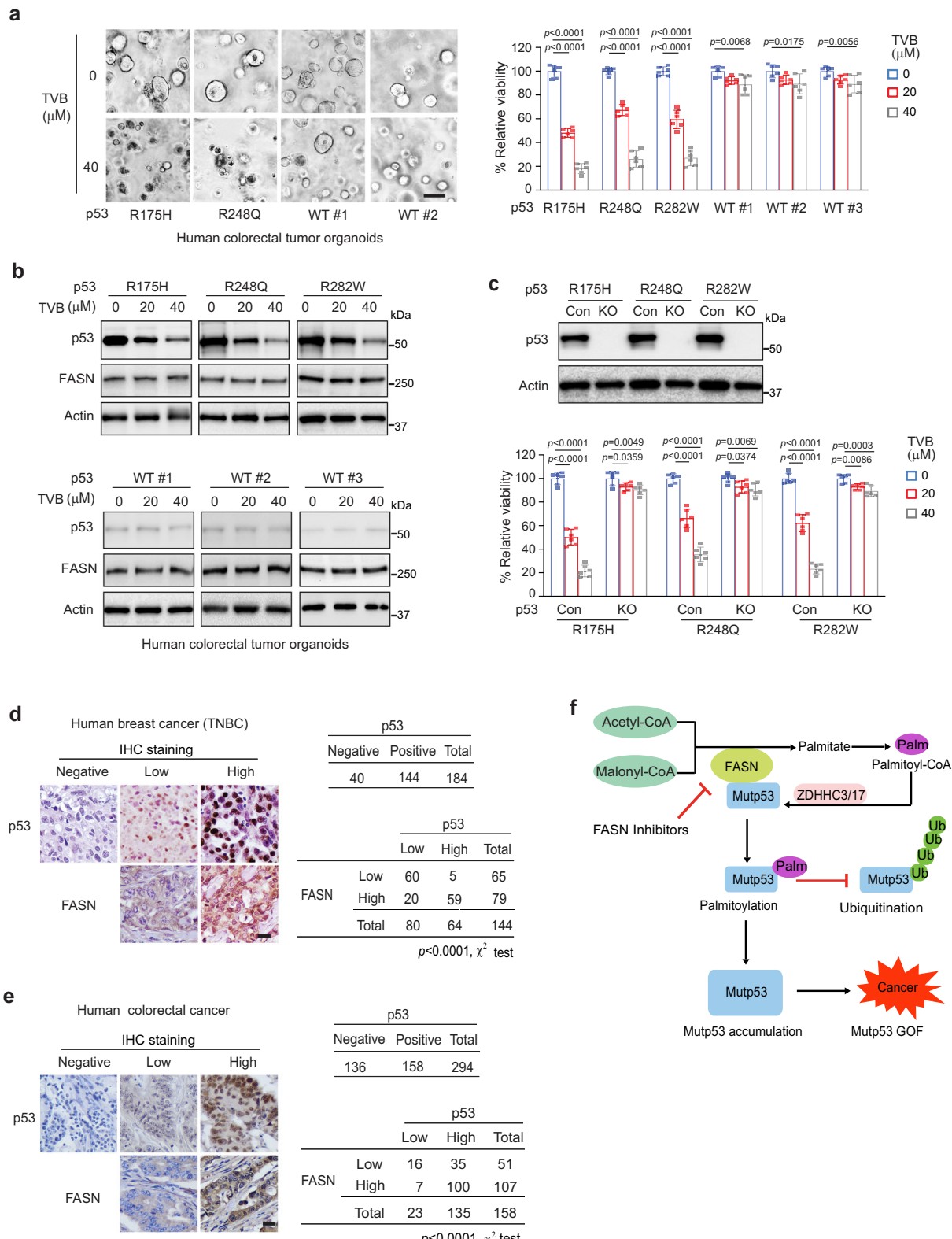

targeted therapies are unavailable for cancer treatment, and the precise mechanisms governing mutp53 accumulation and GOF in cancer cells are not well understood. It has been reported that GOF mutp53 often exhibits altered interactions with other proteins, dysregulated gene expression patterns, and aberrant signaling pathways in cancer cells, which collectively contribute to cancer progression, including promoting cell proliferation, migration and metastasis, metabolic reprogramming, and resistance to therapies[3–6]. Further, different forms of mutp53 often differ in the magnitudes of their GOF activities and display GOF activities in a highly cell- or tissue-specific and context-dependent manner[5,56,57]. Further understanding the mechanism underlying mutp53 accumulation and GOF is crucial for the successful development of mutp53-targeted therapies. In this study, we found that FASN specifically interacted with mutp53 and promoted

**Fig. 7 | Pharmacological inhibition of FASN preferentially suppresses human colorectal tumor organoids expressing mutp53. a** TVB-3166 treatment displayed a more pronounced inhibitory effect on human colorectal tumor organoids expressing mutp53 compared with organoids expressing wtp53. Left panels: Representative images of different organoids expressing mutp53 or wtp53, which were treated with or without TVB-3166 for 48 h. Scale bar: 50 μm. Right panel: The relative viability of mutp53 and wtp53 organoids. **b** TVB-3166 reduced the protein levels of mutp53, but not wtp53, in organoids. **c** TVB-3166 treatment displayed a more pronounced inhibitory effect on colorectal tumor organoids expressing mutp53 compared with organoids with mutp53 KO by CRISPR/Cas9. Upper panel: Western blot analysis of mutp53 levels in organoids with or without mutp53 KO. Lower panel: The relative viability of organoids with or without mutp53 KO. In

**a, c** data represent mean ± SD ($n = 6$ independent experiments). One-way ANOVA followed by Dunnett's test. **d, e** FASN expression is positively associated with mutp53 levels in human breast and colorectal cancer specimens. FASN and p53 levels were analyzed by IHC staining on TMAs. Left: Representative IHC staining images. Right: Summary of IHC staining results. Only tumor specimens with positive p53 staining (mutp53) were used for the association analysis (left lower). Scale bars: 20 μm. Two-sided $\chi^2$ test. **f** A schematic model illustrating that FASN specifically interacts with mutp53, enhances mutp53 palmitoylation and inhibits its ubiquitination to promote mutp53 protein accumulation and GOF in tumorigenesis. Palm: palmitoyl-CoA. In **b, c**: Western blot data represent three repeats with similar results. Source data are provided as a Source Data file.

mutp53 palmitoylation, which in turn inhibited mutp53 ubiquitination and degradation to promote mutp53 accumulation and GOF. Notably, targeting FASN genetically or pharmacologically suppressed mutp53 palmitoylation, accumulation and GOF in tumorigenesis. These results suggest that FASN-mediated mutp53 palmitoylation is an important mechanism underlying mutp53 accumulation and GOF, and most importantly, targeting FASN is a potential therapeutic strategy for cancers carrying mutp53 (Fig. 7f). In line with our results, a previous study showed that cell lines lacking p53 or expressing mutp53 exhibited increased sensitivity to apoptosis induced by FASN inhibition[58]. Interestingly, it was reported that FASN in *C. elegans* is a target gene of CEP-1, the p53 homolog in *C. elegans*, and the human FASN gene is a target of p53 family members TAp73α and ΔNp63α but not p53, suggesting an interesting but complex link between FASN and the p53 family members[59].

It has been well-known that the mechanisms for the regulation and modifications of wtp53 and mutp53 are very different and even same regulation and modifications can lead to different biological consequences on wtp53 and mutp53[3,5,6]. Recently, wtp53 was reported to be palmitoylated by ZDHHC1, which promoted the nuclear localization but not stability of wtp53 protein[46]. Furthermore, wtp53 palmitoylation by ZDHHC1 enhanced tumor-suppressive function of wtp53 in s.c. xenograft tumor models[46]. In this study, we found that the regulatory mechanisms and biological consequences of palmitoylation are very different between wtp53 and mutp53. We found that mutp53 exhibited a much higher level of palmitoylation than wtp53, suggesting that palmitoylation is a more critical post-translational modification for mutp53 than for wtp53. Further, FASN directly interacted with mutp53, but not wtp53, to promote mutp53 palmitoylation. FASN-mutp53 direct interaction is expected to lead to a higher local concentration of palmitate in close proximity to mutp53. Indeed, FASN displayed a critical role in palmitoylation of mutp53 but not wtp53. We further found that ZDHHC3/17, but not ZDHHC1, played a major role in catalyzing mutp53 palmitoylation. Interestingly, unlike wtp53, palmitoylation did not affect mutp53 localization but promoted mutp53 stability and accumulation. While wtp53 palmitoylation promoted the tumor-suppressive function of wtp53[46], mutp53 palmitoylation enhanced mutp53 GOF to promote tumorigenesis. Importantly, blocking FASN dramatically suppressed mutp53 palmitoylation, accumulation and GOF, which in turn exhibited a more pronounced inhibitory effect on the growth of mutp53 tumors than wtp53 or $p53^{-/-}$ tumors. Currently, the mechanism leading to these obvious differences remains unclear. Missense mutations were reported to result in p53 protein conformation alterations[3–5], which may affect the accessibility of different proteins including FASN and palmitoyltransferases to p53, leading to different protein interaction patterns between mutp53 and wtp53. However, a significant amount of our current knowledge on the mutp53 conformation changes came from the studies using antibodies against different forms of mutp53, including the concept of two major categories of mutp53, conformational mutants (e.g. R175H) and contact mutants (e.g. R273H)[60]. Interestingly, a recent study using extensive all-atom molecular dynamics simulations

showed that both R175H and R273H mutp53 displayed different structures from wtp53[61]. It has been reported that many different sites in p53 can undergo different types of modifications, and the crosstalk between these modifications can impact the modifications, confirmations and functions of p53[62]. In addition to our finding in this study that mutp53 palmitoylation suppresses its ubiquitination, palmitoylation was also reported to modulate nitrosylation of substrate proteins[63]. Interestingly, glutathionylation of wtp53 was reported to inhibit DNA binding function of wtp53[64]. These findings suggest that other types of post-translation modifications of wtp53 may affect the accessibility of FASN and palmitoyltransferases to wtp53. Future studies using comparative analyses of RNAseq, proteomics, and metabolomics are needed to reveal the mechanisms underlying these differences.

While palmitate is the direct product of FASN, interestingly, our results showed that exogenous palmitate has a moderate effect on mutp53 palmitoylation and accumulation and cannot fully rescue the decreased mutp53 palmitoylation and protein levels resulting from FASN KO in cultured cells (Supplementary Fig. 6b-e). Currently, it is unclear whether dietary fatty acid supplementation or a high-fat diet can effectively rescue the decreased mutp53 palmitoylation and impaired GOF in tumorigenesis resulting from FASN deficiency or inhibition in animal models. Interestingly, several recent studies reported that the dietary fatty acid supplementation or high-fat diet alone cannot effectively reverse the phenotypes resulted by FASN inhibition in mouse models. For instance, exogenous palmitate supplementation failed to prevent vision defects resulting from FASN KO in mice, suggesting the de novo lipogenesis-dependent pathways intrinsic to the eye are crucial for vision[65]. FASN was reported to be crucial for accurate myelination and myelin growth, and dietary lipid intake only partially compensated the defects caused by FASN deficiency in mice[66]. Additionally, FASN deletion in macrophages prevented diet-induced insulin resistance, macrophage recruitment to adipose tissue, and chronic inflammation in mice, which cannot be rescued by exogenous palmitate[67]. These findings suggest that locally and newly synthesized palmitate through the FASN-mediated de novo lipogenesis other than palmitate from extracellular sources plays a crucial role in regulation of some critical cellular functions[65,68]. Future studies should investigate the effect of dietary fatty acid supplementation and a high-fat diet on mutp53 palmitoylation and GOF in animal models, which has the potential impact on the intervention and treatment of cancers carrying mutp53.

## Methods

### Cell lines, vectors, and reagents

SK-BR3, MDA-MB468, and LS1034 cell lines were provided by the American Type Culture Collection (ATCC). The isogenic $p53^{+/+}$, $p53^{-/-}$ and $p53^{R248W/-}$ HCT116 cell lines were generously provided by Dr. Bert Vogelstein (Johns Hopkins University). Mouse embryonic fibroblasts (MEFs) were isolated from E13.5 embryos of $p53^{+/+}$, $p53^{-/-}$ and $p53^{R172H/R172H}$ mice and subsequently transduced with pBabe-E1A and pBabe-RasV12 retroviral vectors (Addgene) to generate transformed MEFs. The pLPCX vectors expressing different GOF mutp53 were generated via PCR

amplification. The site-directed mutagenesis kit (QuikChange II XL, Agilent Technologies) was used to mutate cysteine residues in pLPCX-R175H p53 vectors. The pcDNA3.1-FASN vector (Addgene) was used as a template to reconstruct the retroviral pLPCX-FASN-Flag vector. shRNA vectors for FASN and p53 were constructed by inserting target sequences into the PLKO.1 hygro lentiviral shRNA vector (Addgene), with specific sequences presented in Supplementary Table 2. The target sequences for siRNAs against 23 human ZDHHCs have been described previously[47]. Small molecule inhibitors for FASN, TVB-3166 and Orlistat, were purchased from MedChemExpress (Monmouth Junction, NJ). 2-bromopalmitate (2-BP) was purchased from Selleckchem (Houston, TX), and Palmostatin B was purchased from Sigma-Aldrich (St. Louis, MO).

### Establishing the mutp53 and/or FASN KO cell lines using the CRISPR/Cas9 system
The mutp53 and/or FASN KO cell lines were generated as previously described[69]. In brief, the sgRNAs for *p53* and *FASN* were designed using an online CRISPR tool[69], and sgRNA sequences were inserted into the lentiCRISPRv2-blast or lentiCRISPRv2-puro vectors (Addgene). The sgRNA sequences are listed in Supplementary Table 2. Blasticidin- or puromycin-resistant cell colonies were selected. The deletions of target genes were verified through PCR amplification of the edited regions followed by sequencing, as well as by Western blot analysis of p53 and/or FASN proteins.

### Western blot assays
Total protein was extracted from cells and tissues using RIPA buffer (Thermo Fisher Scientific) supplemented with protease inhibitors, and protein concentrations were determined using the Bradford reagent. The lysates were separated by SDS-PAGE gels and then transferred onto PVDF membranes. The following antibodies were used for assays: anti-Flag-M2 (F1804, Sigma-Aldrich; 1:20,000 dilution), anti-β-Actin (A5441, Sigma-Aldrich; 1:10,000 dilution), anti-HA (3F10, Roche; 1:1000 dilution), anti-His (sc803, Santa Cruz Biotechnology; 1:1000 dilution), anti-GST (sc-138, Santa Cruz Biotechnology, 1:5000 dilution), anti-FASN (sc55580, Santa Cruz Biotechnology; 1:3000 dilution), anti-Lamin A/C antibody (sc-7292, Santa Cruz Biotechnology; 1:5000 dilution), anti-GAPDH (sc-47724, Santa Cruz Biotechnology; 1:5000 dilution), anti-mouse p53 (Leica Biosystems, CM5; 1:2000 dilution), and anti-human p53 (sc126, Santa Cruz Biotechnology; 1:2000 dilution) antibodies. Western blot intensity quantification was conducted using ImageJ software (NIH).

### Co-IP and LC−MS/MS
For co-IP of FASN-Flag and mutp53 proteins in cells with ectopic expression of these proteins, the anti-p53 (sc126AC, Santa Cruz Biotechnology) and anti-Flag (A2220, Sigma-Aldrich) agarose beads were used to pull down mutp53 and FASN-Flag in cell lysates, respectively. To identify the interaction domains between FASN and mutp53, the anti-Flag (M2) and anti-HA (HA-7, Sigma-Aldrich) agarose beads were utilized to immunoprecipitate FASN-Flag and HA-p53, respectively. The eluted proteins were subsequently analyzed by Western blot. For co-IP of endogenous FASN and p53, the anti-FASN antibody (sc55580, Santa Cruz Biotechnology) and anti-p53 antibody (sc126, Santa Cruz Biotechnology) were employed, with mouse IgG serving as a negative control. To screen for specific mutp53-binding proteins, *p53*[−/−] HCT116 cells transduced with empty control vectors or vectors expressing R175H mutp53 or wtp53 were employed for co-IP by using the anti-p53 beads (sc126AC, Santa Cruz Biotechnology). The p53 protein complex was eluted by 0.1 M glycine solution, separated by SDS-PAGE gels, and visualized by silver staining using a silver staining kit (Invitrogen). The eluted proteins were analyzed by LC-MS/MS at the Biological Mass Spectrometry facility of Rutgers University. LC-MS/MS was performed using a nano LC (Dionex Ultimate 3000 RLSCnano System; Thermo-Fisher) interfaced with an Eclipse Tribrid mass spectrometer (ThermoFisher)[70]. Peak list MASCOT Generic Format (MGF) files were generated by Thermo Proteome Discoverer (v.3.0) and searched against Uniprot human reference proteome and a database composed of common lab contaminants (CRAP) using an in-house installation of GPM Fury[71] (X!Tandem Alanine). Search parameters are as follows: fragment mass error, ± 20 ppm; parent mass error, ± 7 ppm; fixed modification, carbamidomethylation on cysteine; variable modifications: methionine mono-oxidation for the primary search, asparagine deamination, tryptophan oxidation and dioxidation, methionine dioxidation, and glutamine to pyro-glutamine were considered at the refinement stage. Protease specificity: trypsin (C-terminal of R/K unless followed by P with 1 missed cleavage during the preliminary search and 5 missed cleavages during refinement. Minimum acceptable peptide and protein expectation scores were set at $10^{-2}$ and $10^{-4}$, respectively. The overall peptide false positive rate[72] was 0.07%.

### In vitro His or GST pull-down assays for protein-protein direct interactions
In vitro His and GST pull-down assays were conducted as previously described to analyze the direct interactions between proteins using recombinant proteins[69,73]. Briefly, E. coli (BL21 DE3) carrying pET-32a-His-mutp53 or His-wtp53 was induced with 0.4 mM IPTG at 16 °C for 16 h to express recombinant His-tagged mutp53 (R175H) or wtp53. The purified His-tagged proteins were immobilized on Ni-NTA agarose beads (QIAGEN) and incubated with purified Flag-FASN (TP308644, Origene). His protein alone served as a negative control. After thorough washing, bound proteins were eluted, separated by SDS-PAGE gels, and then analyzed by Western blot assays. For GST pull-down assays[69,73], purified GST-ZDHHC3 and GST-ZDHHC17 proteins were immobilized on Glutathione-Sepharose beads and incubated with purified His-mutp53 (R175H) or His-wtp53 proteins. GST alone was used as a negative control. Following washing, bound proteins were eluted and examined using Western blot assays.

### PLA
The in situ interaction between endogenous FASN and p53 in cells was detected using a Duolink PLA kit (Sigma-Aldrich)[73]. Cells were first fixed and then incubated with primary antibodies targeting FASN (10624-2-AP, Proteintech) and p53 (sc126, Santa Cruz Biotechnology), respectively. Duolink secondary antibodies were subsequently added, and ligation was performed to enable the detection of proteins in close proximity. The nuclei were counterstained with 4′,6-diamidino-2-phenylindole (DAPI). PLA signal quantification was conducted using the ImageJ software (NIH).

### ABE assays
ABE assays were performed as previously described[43]. Briefly, samples were homogenized in the lysis buffer (100 mM Tris-HCl, pH 7.2, 5 mM EDTA, 150 mM NaCl, 2.5% SDS, and the protease inhibitor cocktail (Sigma-Aldrich)) with 50 mM NEM (E3876, Sigma-Aldrich), and were solubilized for 2 h. The supernatants obtained by centrifugation at 16,000 g for 20 min were subjected to chloroform/methanol/water (v/v 1:4:3) to remove excess NEM. The precipitate was then dissolved in 4% SDS buffer. Concentrated proteins were treated with 0.5 M HAM (159417; Sigma-Aldrich) or 0.5 M NaCl for 2 h and then subjected to chloroform/methanol protein precipitation again. Protein was solubilized in 2% SDS buffer, followed by incubation with streptavidin-agarose beads (20353, Thermo Fisher Scientific) for 1.5 h. The immunoprecipitated beads were washed with the lysis buffer and eluted protein samples were subjected to Western blot assays to detect palmitoylated p53 protein using an anti-p53 antibody (sc126, Santa Cruz Biotechnology; 1:2000 dilution).

## Metabolic labeling with 17-ODYA and click chemistry

Assays were performed as described[44]. In brief, cells were incubated with 50 µM 17-ODYA (90270, Cayman Chemical) in culture media for 6 h. Cells were then washed in PBS before chasing in culture media supplemented with 300 µM palmitate, 10 µM Palmostatin B or DMSO (vehicle) added at chase time 0. At indicated time points, cells were collected, washed, and suspended in the lysis buffer (50 mM Tris-HCl, pH 7.5, 150 mM NaCl, 10% glycerol, and 0.2% Triton X-100) containing 10 µM Palmostatin B and 50 mM NEM. Following lysis for 30 min, the lysate was centrifuged, and the resulting supernatants were incubated with anti-p53 beads (sc126AC, Santa Cruz Biotechnology). After incubation, the beads were washed with lysis buffer, followed by Cu(I)-assisted click reactions. For click reactions, the beads were incubated in PBS supplemented with 10 µM IRDye 800 CW azide infrared dye (LI-COR), 1 mM CuSO4, 1 mM TCEP, and 0.1 mM TBTA for 1 h, and then washed with PBS and resuspended in SDS buffer. Samples were then separated by SDS-PAGE and transferred onto PVDF membranes. The fluorescence was detected using the ChemiDoc MP Imaging System (Bio-Rad).

## qPCR assays

Total RNA in cells and tissues was extracted using an RNeasy Kit (QIAGEN), followed by reverse transcription using a TaqMan Reverse Transcription Kit (Applied Biosystems) according to the manufacturer's instructions. qPCR was conducted using the TaqMan PCR Master Mix (Applied Biosystems)[69]. The mRNA levels of genes analyzed were normalized to the mRNA levels of Actin.

## IF staining assays

IF staining was performed as previously described[74]. An anti-p53 antibody (sc126, Santa Cruz Biotechnology) was employed to detect p53 in cells. Nuclei were stained using DAPI. Slides were incubated with the Alexa Fluor 555 Goat Anti-Mouse IgG (H + L) secondary antibody (A28180, Invitrogen). The slides were then observed under a fluorescence microscope.

## Ubiquitination assays

Ubiquitination assays were conducted as described[69]. In brief, cells were transfected with various expression vectors, including those encoding mutant or WT p53, along with HA-tagged ubiquitin (HA-Ub). To inhibit the proteasomal degradation of proteins, cells were treated with the proteasomal inhibitor MG132 (10 µM) for 12 h before being collected for the ubiquitination assays. To assess the ubiquitination status of mutp53 and wtp53, p53 in cell lysates was pulled down using an anti-p53 antibody (sc126, Santa Cruz Biotechnology), followed by Western blot assays using an anti-HA antibody (11867423001, Sigma-Aldrich) to detect HA-Ub.

## CHX-chase assays

CHX-chase assays were performed to determine the half-life of mutp53 protein in cells[69]. Cells were first treated with the protein synthesis inhibitor CHX (50 µg/mL) for various durations (4–16 h) to inhibit new protein synthesis. Control cells were treated with DMSO. Following treatments, cell were collected and subjected to Western blot analysis to measure the mutp53 protein levels in cells, utilizing an anti-p53 antibody (sc126, Santa Cruz Biotechnology).

## Cytosolic and nuclear fractionation

Nuclear and cytosolic fractions were isolated using a Nuclear/Cytosolic Fractionation kit according to the manufacturer's instructions (AKR-171, Cell Biolabs). Briefly, cells were collected and cell pellets were suspended in the Cytosol Extraction Buffer containing DTT and protease inhibitors. After incubation on ice for 10 min, cells were lysed in the Cell Lysis Reagent and vortexed, followed by centrifugation. The resulting supernatants (the cytoplasmic fraction) were collected. The nuclear pellets were lysed in the Nuclear Extraction Buffer on ice for 30 min and vortexed, followed by centrifugation. The resulting supernatants (the nuclear fraction) were collected. Both fractions were used for Western blot analysis.

## AIG assays

To assess AIG ability of cells, cells were plated onto culture dishes pre-coated with a layer of culture media mixed with 0.6% agarose. Cells were seeded on the top layer of culture media containing 0.3% agarose. For inhibitor treatments, the top agarose layer included TVB-3166, Orlistat, or a vehicle control. Fresh inhibitors were added daily for 3 consecutive days. After 2–3 weeks, colonies were stained with crystal violet, and the number of colonies was counted.

## Tumorigenesis assays in mice

All animal studies were conducted with approval from the Rutgers University Institutional Animal Care and Use Committee (IACUC). Mice were housed under controlled conditions, including a temperature of $22 \pm 1\,°C$, 40–60% relative humidity, as well as a 12-hour light/dark cycle. For orthotopic breast tumor models, SK-BR3 cells ($5 \times 10^6$) suspended in a 1:1 mixture of DMEM and Matrigel were injected into the mammary fat pads of 8-week-old female BALB/c athymic nude mice (Taconic; $n = 10$ per group) as described[69,73]. For s.c. xenograft tumor models, $p53^{+/+}$, $p53^{-/-}$ or $p53^{R248W/-}$ HCT116 cells ($5 \times 10^6$ in 0.2 mL of PBS) were injected s.c. into 8-week-old BALB/c athymic nude mice (Taconic; $n = 10$ per group; half male and half female). For inhibitor treatments, once the tumor volume reached approximately 30 mm³, the mice received TVB-3166 via oral gavage (60 mg/kg) every other day for three weeks, while control mice received vehicle. The tumor volume was calculated as 1/2 (length × width²). One day post-treatment, mice were euthanized, and tumors were excised for further assays, including tumor weight measurements. Additionally, C57BL/6 J $p53^{R172H/R172H}$ mice (a generous gift from Dr. Guillermina Lozano at MD Anderson Cancer Center)[54], $p53^{+/+}$, and $p53^{-/-}$ mice (The Jackson Laboratory) were used to assess TVB-3166 effects. Starting at 8 weeks of age, mice received TVB-3166 (60 mg/kg) via oral gavage every other day for 12 weeks ($n \geq 17$ per group; half male and half female). All mice were scheduled for euthanasia once their tumor volume reached 1700 mm³, in accordance with the approved IACUC protocol. The maximal tumor size of all mice used in the present study did not exceed 1,700 mm³.

## IHC assays and scoring

TMAs for human colorectal cancer (CO6161) and breast cancer (BR489, BR1902, and BR1301) were provided by US Biomax, with all samples deidentified. IHC staining was performed using anti-FASN (sc55580, Santa Cruz Biotechnology) and anti-p53 (sc126, Santa Cruz Biotechnology) antibodies to detect FASN and mutp53 in tumor samples, respectively. The staining results were evaluated using a scoring system ranging from 0 to 9. Scores were calculated by multiplying the signal intensity with the percentage of positively stained cells. Signal intensity was scored as: 0 (no signal), 1 (weak), 2 (moderate), and 3 (strong), while the percentage of positive cells was categorized as: 0 (0%), 1 (<25%), 2 (25-50%), and 3 (>50%). For IHC staining of p53 and Ki-67 in xenograft tumors, anti-p53 (sc126, Santa Cruz Biotechnology) and anti-Ki-67 (556003, BD Biosciences) antibodies were used. For Ki-67 IHC staining, six tumor sections were analyzed per group, with five randomly selected fields counted from each section. The percentage of Ki-67 positive cells was calculated by dividing the number of positive cells by the total number of cells, and the average percentage was determined for each group.

## Human organoid culture, viability assessment, and lentivirus transduction

Human colorectal cancer organoids expressing different endogenous mutp53, including HCM-CSHL-0061-C18 (R175H), HCM-CSHL-0244-

C20 (R248Q), and HCM-CSHL-0247-C18-A (R282W), as well as human colorectal cancer organoids expressing endogenous wtp53, including HCM-CSHL-0143-C20 (WT #1), HCM-CSHL-0461-D12 (WT #2), and HCM-CSHL-0376-D37 (WT #3), were provided by ATCC. Organoid viability was assessed after TVB-3166 treatment for 48 h using CellTiter-Glo 3D reagent (G9681, Promega) according to the manufacturer's instructions. Mutp53 in organoids was deleted by the CRISPR/Cas9 system using the lentiCRISPRv2-blast lentiviral infection. In brief, organoids were resuspended in culture media supplemented with Transdux (System Biosciences) and mixed with 10-fold concentrated viral particles. Organoids were then seeded on polymerized Matrigel in a 12-well plate and incubated overnight at 37 °C, followed by the same procedure for subculture. Blasticidin selection was performed over a period of 3-7 days, followed by the evaluation of p53 KO in organoids using Western blot assays.

## Statistical analysis

Data were collected from a minimum of three independent experiments and presented as mean ± standard deviation (SD) as described in the figure legends. Statistical comparisons between two groups were performed using a two-tailed Student's $t$-test. For comparisons involving multiple groups, analysis of variance (ANOVA) was followed by Dunnett's, Tukey's, or Bonferroni's tests. Mouse survival data were analyzed using Kaplan-Meier survival curves, with statistical differences assessed by the log-rank (Mantel-Cox) test. GraphPad Prism 9 was used for all statistical analyses, with significance set at $p < 0.05$.

## Reporting summary

Further information on research design is available in the Nature Portfolio Reporting Summary linked to this article.

## Data availability

Source data are provided with this paper. The data from LC-MS/MS analysis were deposited in the MassIVE repository (http://massive.ucsd.edu; the accession number: MSV000095580). The remaining data are available within the Article, Supplementary Information, and Source Data file. Source data are provided with this paper.

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

## Acknowledgements

We would like to thank Haiyan Zheng at the Biological Mass Spectrometry facility of Rutgers University for assistance with LC-MS/MS proteomic analysis. This work was supported in part by grants from the National Institutes of Health (R01CA214746 and R01CA229257 to Z.F., as well as R01CA203965 and R01CA260837 to W.H.), New Jersey Commission on Cancer Research (to Z.F.), and from the Breast Cancer Research Foundation (to B.G.H.). Jie L. and J.W. were supported in part by the post-doctoral fellowship from New Jersey Commission on Cancer Research.

## Author contributions

J.L., Y.S., Jie L., D.X., C.C., J.W., and J.Z. performed the experiments and analyzed data. J.B., L.Z., and S.D. were involved in data analysis. B.G.H. contributed important materials. W.H. and Z.F. conceived and supervised the study. J. L., W.H., and Z.F. wrote the manuscript.

## Competing interests

The authors declare no competing interests.
