## [Transparent Peer Review file · Nature Communications]

Lipogenic Enzyme FASN Promotes Mutant p53 Accumulation and Gain-of-function through Palmitoylation

Corresponding Author: Professor Zhaohui Feng

Version 0:

Reviewer comments:

Reviewer #1

(Remarks to the Author)

In this manuscript by Liu et al, they reported a functional link between mutated p53 and FASN in which FASN stabilizes mutant p53 through palmitoylation allowing its gain of function in cancer. They first identified interactions between FASN and mutated p53 through LC-MS/MS and subsequent experiments comprehensively showed that this interaction is specific to mutated p53 but not its wild-type counterpart and FASN inhibition could suppress tumor growth in cell culture systems and mouse models. The data presented is important as it demonstrates a novel link between two heavily implicated oncogenes in cancer, which are FASN and mutated p53. This is especially important in providing a strategy in treating cancer patients with GOF mutated p53 with FASN overexpression. While this manuscript is well-written and experiments are well-executed, there are several concerns below that the authors need to address before this manuscript can be considered for publication.

1. Can the authors explain why only 4 of 10 cysteine residues are presented (Fig. 3c). How about the remaining 6 residues? The authors should include them in Supplementary and explain the results.
2. On pg. 10, the word "in vivo" should be removed as it is confusing owing that this usually refers to animal model.
3. Though p53 levels can act as surrogate for mutation positivity in IHC, I feel it is important to include several cancer cases of known p53 status (both wt and mutant) for Fig. 7d and e.
4. Western blot data of FASN for Fig. 2f-h, 3d, 3g, 3j, 3i, 6f, 6j, and 7b to show that changes in p53 levels correlate with FASN.
5. A missing piece of data in this manuscript is rescue data on FASN's role in promoting mutp53 GOF. A suggestion will be to incorporate FASN by plasmid transfection in TVB- or siRNA- treated cells.

Reviewer #2

(Remarks to the Author)

The manuscript by Liu et al. titled: "Lipogenic Enzyme FASN Promotes Mutant p53 Accumulation and Gain-of-function in Cancer through Palmitoylation" reports the finding that the gain of function (GOF) mutant p53 tumor suppressor is palmitoylated preferentially over the wild type p53 and the modification is mediated by fatty acid synthase as well as palmitoyl transferases DHHC3 and DHHC17.

Palmitoylation of mutp53 prevents ubiquitination and degradation of the protein leading to increased levels of mutp53 protein. Inhibition of FASN with existing small molecule inhibitors blocks mutp53 palmitoylation and decreases mutp53 protein levels. Both genetic inhibition of FASN and pharmacologic inhibition reduce mutp53 tumor growth in orthotopic xenograft tumor models. The effect on tumor growth is striking under the conditions tested.

The findings overall are interesting to a broad cancer audience and since existing drugs that inhibit FASN are under clinical studies there is significant translational relevance.

The data are of high quality and well controlled and are one of the strengths of the manuscript. How the multiple players, mutp53, FASN, and DHHC3/17 work together spatially, and the functional role of each component is less clear. DHHC enzymes are 4-pass transmembrane enzymes that localize to different membrane compartments. Since FASN and mutp53 appear to interact in the nucleus (based on the PLA result Fig. 1F) where are DHHC3 and 17 and when do they come into play?

The finding that FASN is essential in regulation of palmitoylation of p53 is intriguing. While FASN related proteins have been shown to be capable of palmitoylating peptides the generally accepted mechanism of protein S-palmitoylation is by DHHC enzyme mediated transfer of palmitate. While the authors provide compelling evidence that blocking FASN blocks palmitoylation it's unclear if the mechanism is simply through depletion of palmitate. Whether FASN is directly palmitoylating mutp53 or if it is indirectly involved is still unclear.

The interaction of FASN with mutp53 over wild type p53 is strikingly selective as is the preferential palmitoylation of mutp53. While the authors narrow down the region of interaction to the DNA binding domain of mutp53 where both the GOF mutations and palmitoylated cysteines reside, it's unclear how these mutations mediate the novel interaction. The role of FASN in this process requires further investigation.

Specific comments:

My main concern is that the conclusion that FASN increases mutp53 palmitoylation may be an overinterpretation of the data albeit a subtle one. The data show FASN is required for high levels of palmitoylated mutp53 and without FASN levels are decreased. Since FASN has not definitively been shown to transfer palmitate to mutp53 or any protein rigorously it is an assumption that the presence of active FASN increases palmitoylation as opposed to reduces the rate of depalmitoylation. Multiple enzymes have been identified that remove palmitate from proteins. It's possible the interaction of FASN with mutp53 requires palmitoylation of mutp53 by DHHC enzymes and this interaction with FASN prevents removal of palmitate by thioesterases.

Is mutp53 palmitoylation still reduced upon FASN inhibition/KD if cells are treated with APT1 inhibitor palmostatin B?

The p53-FASN PLA assay shows the two proteins interact in the area of the nucleus, this should be quantified. Can the authors determine if the puncta are inside or outside the nuclear envelope? FASN is generally in the cytoplasm, a western blot of the nuclear and cytoplasmic fractions should be included to show the level of FASN in the nucleus. Nuclear FASN has been reported in prostate cancer (Madigan et al Am J Pathol. 2014 Aug;184(8):2156-62), but I am not aware if this is generally true in other cancer cells.

Twenty-four hours is a relatively long time to inhibit FASN and could have broad effects. A time course of pharmacologic inhibition of FASN may help clarify if the reduction in palmitoylation is caused by general depletion in palmitate (only after 24 hours) or a local decrease or if FASN directly palmitoylates mutp53 (short time points).

Does FASN inhibition for 24 hours reduce DHHC3 and 17 auto-palmitoylation (by ABE)? This would suggest that palmitate is limiting at this time point.

Does FASN inhibition prevent FASN interaction with mutp53?

Conversely does FASN interact with mutp53 containing the cysteine mutations or when DHHC3/17 are knocked down? This would help determine if FASN binding requires the palmitoylated residues on mutp53 and binding is after palmitoylation.

Some molecular characterization of the treated and untreated tumors in addition to the p53 levels presented would strengthen the conclusions. It would be informative if the transcriptional changes known to occur in response to GOF mutp53 are altered in the tumors treated with the FASN inhibitors and if they are consistent with tumor growth inhibition. This doesn't need to be global analysis, just some of the known target genes. Is proliferation of the tumor cells reduced or is cell death increased?

Minor comments.

I think general readers are less familiar with the gain of function mutations in p53. Can the authors elaborate on how frequent the GOF mutations are in different cancers?

It would be informative as to why was FASN chosen for further study out of the proteomic interaction screen over other potential candidates.

Reviewer #3

(Remarks to the Author)

In the article 'Lipogenic enzyme FASN promotes mutant p53 accumulation and gain-of-function in cancer through palmitoylation' the authors characterise a role for FASN in preventing degradation of mutant p53 proteins. The article is well-written and logically structured. Experiments are well-controlled and data are interpretable.

Major comments:

1. In figure 3, the authors show that 4 cysteines in p53 are important for binding to FASN and palmitoylation of p53. Do the

authors also see a reduced binding of ZDHHC3/17 to this mutant?

2. One unresolved question is why FASN and the ZDHHCs bind various mutants, but not WT p53. The authors speculate in the discussion that this could be to do with the folding status of the mutants/ accessibility to p53. Some of the mutants that were analysed are presumed to still normally fold. Does any other modification that occurs on cysteines in WT p53 prevent palmitoylation? Maybe glutathionylation? Or does DNA binding of WT p53 prevent access to these cysteines.

3. It would be good if the authors could put the current findings in the context of other literature around p53 and FASN. E.g. Li et al 2001 PMID: 11245456 and D'Erchia 2006 PMID: 16582625. Especially the first article already describes the usefulness of FASN inhibitors in cells that have no functional p53.

4. Do the authors see a difference in numbers of tumours/aggressiveness of the tumours in the mice used in figure 6K? Do tumours in the TVB treated mice have lower palmitoylation and lower 172H levels?

Reviewer #4

(Remarks to the Author)

Liu et al, present an interesting study on the occurrence and function of palmitoylation in the gain-of-function(GOF) mutant of p53. The authors suggest that the p53GOF mutant physically interacts with FASN and undergoes palmitoylation catalyzed by ZDHHC3 and ZDHHC17. While the palmitoylation-mediated regulation of wild-type p53 has been documented before, this study highlights distinctions in the mechanism and functional consequences of palmitoylation in the context of the p53 GOF mutation. This work is clearly explained, and the experiments, conducted across various cell lines, seem robust and their outcome convincing. Nonetheless, there are a number of points that require attention.

1. It remains unclear why and how all distinct GOF mutations facilitate binding to FASN, ZDHHC3, and ZDHHC17, leading to palmitoylation, whereas this phenomenon does not occur in wild-type (wt) p53. Do various GOF mutations induce comparable structural changes to the protein? Do the diverse p53 GOF mutants elicit comparable alterations to the cellular state? What determines the differential binding and regulation of palmitoylation among these mutants?

2. Building on the previous point, do p53 gain-of-function(GOF) mutations establish a comparable cellular environment that facilitates or promotes the palmitoylation process? This question could be addressed through comparative analyses of RNAseq, proteomics, and/or metabolomics.

3. What is the contribution of variable levels of palmitate to WT and GOF p53 palmitoylation? Despite previous descriptions of FASN acetyltransferase activity, the current study reveals that knockdown of ZDHHC3 and ZDHHC17 prevents palmitoylation of p53 mutants, suggesting that FASN does not fulfill this role. Can elevated palmitate levels overcome the absence of FASN? The authors should also discuss the potential impact of factors such as, for instance a high-fat diet, in the regulation of WT and GOF mutated p53 in cancer.

4. FASN deletion has, as expected a strong effect on tumour development/growth. Although the effect seems more pronounced in GOFp53 tumours, it is also the case that those tumours grow more/faster which entitles larger demands of fatty acids and other (anabolic) metabolites. Therefore, it is misleading to interpret that the primary function of FASN is mediated through GOF p53 palmitoylation. Can the deletion of FASN, along with the simultaneous overexpression of stabilized p53, rescue tumor or organoid growth and development?

Minor

5. Regarding the title:

'Lipogenic Enzyme FASN Promotes Mutant p53 Accumulation and Gain-of-function in Cancer through Palmitoylation'
Consider to remove 'in cancer' as the above seems also to occur in wt cells (MEFs) if p53mut GOF is introduced?

6. Table 1: Potential mutp53-interacting proteins identified by co-IP followed by LC-MS/MS analysis

All interactions are higher for mutp53 when compared to wt p53. How was the data corrected/normalized? Were comparable levels of p53 pulled down? Did bona fide interactors of wt and GOF p53 appear in the pulldowns? Do different GOF mutants present comparable interactomes (related to point 1).

7. The quality of fluorescent images across the manuscript seems to be low and oversaturated.

Version 1:

Reviewer comments:

Reviewer #1

(Remarks to the Author)

The authors have addressed the concerns raised adequately and this is reflected in the significantly improved manuscript. I recommend the manuscript for publication.

Reviewer #2

(Remarks to the Author)

The author's have addressed most of my concerns and have ruled out that FASN is preventing depalmitoylation of mutp53 and confirmed the presence of FASN in the nucleus of the cell lines used. They have quantified the PLA study and clarified that the interaction between FASN and mutp53 is predominately in the nucleus. They have confirmed that DHHC mediated palmitoylation of mutp53 does not mediate the interaction with FASN. They have shown ZDHHC3/17 interacts with mutp53 in the nucleus by PLA and by western blotting shown that ZDHHC3/17 is in the nucleus.

The authors removed the statement that FASN transfers palmitate to mutp53 and appear to favor the mechanism where FASN provides palmitate through de novo lipogenesis for DHHC mediated transfer to mutp53. They have presented strong evidence for this mechanism even though exogenous palmitate only minimally rescued mutp53 palmitoylation in the FASN KO cells.

The author states: These results showed that extracellular palmitate has a mild and limited effect on mutp53 palmitoylation and mutp53 protein levels, which cannot fully rescue the decreased mutp53 palmitoylation and protein levels induced by FASN KO in cells. These new results also suggest that the palmitate produced via FASN-mediated de novo lipogenesis appears to play a more important role in regulating mutp53 palmitoylation and protein levels than exogenous palmitate that affects the general levels of palmitate in cells.

It is important to note that the product of FASN is palmitoyl-CoA the substrate for the DHHC enzymes, not palmitate and this should be added to the schematic in Fig. 7F. Palmitate needs to be converted to palmitoyl-CoA when provided exogenously and could explain the weak effect on rescue. It is possible that palmitoyl-CoA is low in the nucleus and resides in other membrane compartments. However, this should also be true for the click analog 17-ODYA used for labeling.

Why is the incorporation of 17-ODYA into mutp53 after 2 hours incubation equivalent between control HECT116 and HECT116 FASN KO cells in Supp Fig 7a while exogenous palmitate treatment after 24hours in the SK-BR3 (R175H) FASN KO cells have drastically reduced palmitoylation compared to control cells? If exogenous palmitate is less efficient at palmitoylating mutp53 than de novo lipogenesis I would expect a similar reduction in labeling with the 17-ODYA. Did the authors try the palmitate rescue experiment in the HECT116 cells? It might clarify what is happening with the exogenous palmitate experiment if these experiments were performed in the same cell lines. Is there a difference in the cell culture conditions between the click experiment and the palmitate rescue that could explain this difference? This should be clarified before publication especially if this is caused by a cell line specific effect related to conversion of palmitate to palmitoyl-CoA.

Reviewer #3

(Remarks to the Author)

The authors have very thoroughly and successfully addressed all my concerns

Reviewer #4

(Remarks to the Author)

The authors have clearly addressed the points I raised. Thanks!

Minor:

Supplementary Table 1: Potential mutp53-interacting proteins identified by co-IP followed by LC-MS/MS analysis.

Please extend the table to all found proteins in the pulldowns, and upload processed and raw data to public repositories.

Version 3:

Reviewer comments:

Reviewer #2

(Remarks to the Author)

The authors have addressed all of my concerns. Thanks!

With regards to the authors addressing the request by another reviewer to present an longer list of MS identified proteins.

The overall list looks good, but the table provided is shorter than the excel spreadsheet and they should indicate how the list was curated. It looks like it is only proteins that are enriched with the mutant p53.

The link to access the MS data does not work for me and I was unable to review it.

Point-by-point response to Reviewers' comments

Dear Expert Reviewers:

First, we would like to thank you all very much for taking your precious time to review our manuscript and providing positive, extremely insightful and constructive comments on our manuscript. We have performed experiments as you suggested during the past five months and have carefully revised our manuscript accordingly. The new results from suggested experiments and appropriate changes have been incorporated into our revised manuscript. Please see the following our point-by-point response to your comments.

Reviewer #1's comments:

In this manuscript by Liu et al, they reported a functional link between mutated p53 and FASN in which FASN stabilizes mutant p53 through palmitoylation allowing its gain of function in cancer. They first identified interactions between FASN and mutated p53 through LC-MS/MS and subsequent experiments comprehensively showed that this interaction is specific to mutated p53 but not its wild-type counterpart and FASN inhibition could suppress tumor growth in cell culture systems and mouse models. The data presented is important as it demonstrates a novel link between two heavily implicated oncogenes in cancer, which are FASN and mutated p53. This is especially important in providing a strategy in treating cancer patients with GOF mutated p53 with FASN overexpression. While this manuscript is well-written and experiments are well-executed, there are several concerns below that the authors need to address before this manuscript can be considered for publication.

Comment 1. Can the authors explain why only 4 of 10 cysteine residues are presented (Fig. 3c). How about the remaining 6 residues? The authors should include them in Supplementary and explain the results.

Response: Thanks very much for the kind comments and great suggestions!

It is a great suggestion to show results of all 10 cysteine residues. As suggested, we have presented the results for the remaining 6 cysteine residues in **Supplementary Fig. 8a**. Compared with the R175H mutant p53 (mutp53), mutations of these remaining 6 cysteines individually did not clearly affect the palmitoylation levels of mutp53, suggesting that they are not the major palmitoylation sites for mutp53.

Comment 2. On pg. 10, the word "in vivo" should be removed as it is confusing owing that this usually refers to animal model.

Response: Thanks for this good suggestion. As suggested, we have removed the word "in vivo" on Page 10 (Page 12 in this revised manuscript).

Comment 3. Though p53 levels can act as surrogate for mutation positivity in IHC, I feel it is important to include several cancer cases of known p53 status (both wt and mutant) for Fig. 7d and e.

Response: Thanks for this great suggestion. We totally agreed with the reviewer that it is important to include several cancer cases of known p53 status (both wt and mutant) for Fig. 7d and e. In the TMAs that we used (purchased from US Biomax), the p53 status of tumor samples was not provided by the company. Therefore, we performed IHC staining for p53 in several human colorectal tumor specimens with known p53 mutation status (both wt and mutant) that we have collected in our previous reports (Ref 16). IHC images for these colorectal tumor samples expressing wtp53 or mutp53 (R175H and R273H) are presented in **Supplementary Fig. 20**. As shown, tumor samples expressing wtp53 showed negative IHC staining of p53, whereas tumor samples expressing these two mutp53 showed positive IHC staining of p53.

Comment 4. Western blot data of FASN for Fig. 2f-h, 3d, 3g, 3j, 3i, 6f, 6j, and 7b to show that changes in p53 levels correlate with FASN.

Response: Thanks! As suggested, we have included the Western blot data of FASN levels in all these figure panels. Consistent with previous reports (Jafari, N. *et al. Mol Cancer Res.* 2019;17(1):140-152; Aquino, I. *et al. Arch Oral Biol.* 2020; **113**:104707; Zaytseva, Y. *et al. Oncotarget.* 2018; 9(37):24787), the FASN inhibitor TVB-3166 that specifically targets FASN enzyme activity in fatty acid synthesis did not clearly affect FASN protein levels in these cells. Also consistent with previous reports (Ref 39, and Shueng P. *et al. Int J Mol Sci.* 2022; 23(12):6501), FASN inhibitor Olistat that targets FASN enzyme activity in fatty acid synthesis did not clearly affect FASN protein levels in these cells.

Comment 5. A missing piece of data in this manuscript is rescue data on FASN's role in promoting mutp53 GOF. A suggestion will be to incorporate FASN by plasmid transfection in TVB- or siRNA-treated cells.

Response: Thanks for this great suggestion. As suggested, SK-BR3 cells with FASN knockout (KO) by CRISPR/Cas9 were stably transduced with the FASN expression vector to increase the level of FASN protein to a comparable level of endogenous FASN in control SK-BR3 cells. As shown in **Supplementary Fig. 14**, ectopic expression of FASN in SK-BR3 cells with KO of endogenous FASN rescued the downregulation of mutp53 protein levels (**Supplementary Fig. 14a**) and impaired anchorage-independent growth (AIG) in soft agar of SK-BR3 cells resulted from FASN KO (**Supplementary Fig. 14b**). Furthermore, ectopic expression of FASN in the SK-BR3 cells with FASN KO rescued the impaired growth of orthotopic breast tumors resulted from FASN KO (**Supplementary Fig. 14c**). In contrast, in SK-BR3 cells with KO of both FASN and mutp53, restoration of FASN expression by vectors displayed a much less pronounced effect on AIG and growth of orthotopic tumors, compared with cells with FASN KO only (**Supplementary Fig. 14a-c**). These results suggest that in addition to promoting lipid synthesis, promoting mutp53

stabilization and GOF is another important mechanism by which FASN contributes to tumorigenesis.

Reviewer #2's comments:

The manuscript by Liu et al. titled: "Lipogenic Enzyme FASN Promotes Mutant p53 Accumulation and Gain-of-function in Cancer through Palmitoylation" reports the finding that the gain of function (GOF) mutant p53 tumor suppressor is palmitoylated preferentially over the wild type p53 and the modification is mediated by fatty acid synthase as well as palmitoyl transferases DHHC3 and DHHC17.

Palmitoylation of mutp53 prevents ubiquitination and degradation of the protein leading to increased levels of mutp53 protein. Inhibition of FASN with existing small molecule inhibitors blocks mutp53 palmitoylation and decreases mutp53 protein levels. Both genetic inhibition of FASN and pharmacologic inhibition reduce mutp53 tumor growth in orthotopic xenograft tumor models. The effect on tumor growth is striking under the conditions tested.

The findings overall are interesting to a broad cancer audience and since existing drugs that inhibit FASN are under clinical studies there is significant translational relevance.

Comment 1: The data are of high quality and well controlled and are one of the strengths of the manuscript. How the multiple players, mutp53, FASN, and DHHC3/17 work together spatially, and the functional role of each component is less clear. DHHC enzymes are 4-pass transmembrane enzymes that localize to different membrane compartments. Since FASN and mutp53 appear to interact in the nucleus (based on the PLA result Fig. 1F) where are DHHC3 and 17 and when do they come into play?

Response: Thanks very much for the kind comments and great suggestions!

Thanks for raising this good question. As suggested, we quantified the mutp53-FASN interaction signals in cells detected by PLA assays. As shown in **Fig 1f**, the mutp53-FASN interaction signals were detected by PLA assays in both cytoplasmic and nuclear compartments with more interaction signals detected in the nuclear compartment. To further address this question, we have done following new experiments as suggested. Our newly added results from cytoplasmic/nuclear fractionation followed by Western blot analysis showed the presence of FASN in the nuclear fraction although more FASN proteins were in the cytoplasmic fraction in different cell lines (**Supplementary Fig. 1**). Consistent with our observations, FASN was previously reported to exhibit nuclear localization in prostate cancer cells (Ref 34). As suggested, we also determined the subcellular localization of ZDHHC3/17 by cytoplasmic/nuclear fractionation followed by Western blot analysis, and found that ZDHHC3 and ZDHHC17 were detected in both cytoplasmic and nuclear fractions although more ZDHHC3 and ZDHHC17 proteins were detected in the cytoplasmic fraction (**Supplementary Fig. 12a**). PLA assays also confirmed that mutp53 and ZDHHC3/17 interactions were detected in both cytoplasmic and nuclear compartments with more interactions detected in the nuclear compartment (**Supplementary Fig. 12b**).

Comment 2: The finding that FASN is essential in regulation of palmitoylation of p53 is intriguing. While FASN related proteins have been shown to be capable of palmitoylating peptides the generally accepted mechanism of protein S-palmitoylation is by DHHC enzyme mediated transfer of palmitate. While the authors provide compelling evidence that blocking FASN blocks palmitoylation it's unclear if the mechanism is simply through depletion of palmitate. Whether FASN is directly palmitoylating mutp53 or if is indirectly involved is still unclear.

The interaction of FASN with mutp53 over wild type p53 is strikingly selective as is the preferential palmitoylation of mutp53. While the authors narrow down the region of interaction to the DNA binding domain of mutp53 where both the GOF mutations and palmitoylated cysteines reside, it's unclear how these mutations mediate the novel interaction. The role of FASN in this process requires further investigation.

Specific comments:

My main concern is that the conclusion that FASN increases mutp53 palmitoylation may be an overinterpretation of the data albeit a subtle one. The data show FASN is required for high levels of palmitoylated mutp53 and without FASN levels are decreased. Since FASN has not definitively been shown to transfer palmitate to mutp53 or any protein rigorously it is an assumption that the presence of active FASN increases palmitoylation as opposed to reduces the rate of depalmitoylation. Multiple enzymes have been identified that remove palmitate from proteins. It's possible the interaction of FASN with mutp53 requires palmitoylation of mutp53 by DHHC enzymes and this interaction with FASN prevents removal of palmitate by thioesterases.

Is mutp53 palmitoylation still reduced upon FASN inhibition/KD if cells are treated with APT1 inhibitor palmostatin B?

Response: Thanks for raising these important points regarding mutp53 palmitoylation. While our results demonstrate that FASN is necessary for maintaining high levels of palmitoylated mutp53 in cells, we totally agree with the reviewer that there is currently no direct evidence showing that FASN transfers palmitate to mutp53 or any proteins. Therefore, we have deleted this statement in the manuscript.

Our results showed a much higher level of palmitoylation in mutp53 compared to wtp53, suggesting that palmitoylation is a more important post-translational modification for mutp53 than wtp53. Furthermore, we observed a direct interaction between FASN and mutp53, but not wtp53, which suggests an increased local concentration of palmitate in close proximity to mutp53 may play an important role in the palmitoylation of mutp53 (but not wtp53). Therefore, we speculate that the high local concentration of palmitate produced by FASN is crucial for the palmitoylation of mutp53. To further support our hypothesis, we have performed the following new experiments as suggested by this Reviewer and other Reviewers.

We did experiments based on the wonderful suggestion in Comment 4 raised by the Reviewer that a time course of pharmacologic inhibition of FASN may help clarify if the reduction in palmitoylation is caused by general depletion in palmitate (only after 24 hours) or a local decrease or if FASN directly palmitoylates mutp53 (short time points). As suggested, we conducted a time-

course study for TVB-3166 treatment (0, 8, 16, and 24h) to monitor mutp53 palmitoylation, along with ZDHHC3/17 autopalmitylation as a reference. Notably, TVB-3166 treatment induced a much faster decrease in mutp53 palmitoylation compared with ZDHHC3/17 autopalmitylation (**Supplementary Fig. 6a**). At the same time, although at 24 h both ZDHHC3 and ZDHHC17 showed mild reductions of autopalmitylation, mutp53 showed a much more pronounced reduction of palmitoylation compared with ZDHHC3 and ZDHHC17 (**Supplementary Fig. 6a**). This result suggests that the reduction in palmitoylation of mutp53 appears to be caused by a local decrease of palmitate through the FASN-mediated de novo lipogenesis, which supports the hypothesis that FASN interacts with mutp53 to directly regulate mutp53 palmitoylation.

Furthermore, as suggested by Reviewer 4 (Comment 3), we treated cells with exogenous palmitate, the direct product of FASN that is used for palmitoylation modification, to investigate whether simply increasing palmitate levels in cells can increase mutp53 palmitoylation and protein levels and rescue the decreased mutp53 palmitoylation and protein levels resulted from FASN KO. Interestingly, treating cells with different concentrations of exogenous palmitate (30 and 60 μM for 24 h) resulted in a mild increase in mutp53 and wtp53 palmitoylation in cells (**Supplementary Fig. 6b, c**). Notably, palmitate treatment only slightly but did not fully rescue the decreased mutp53 palmitoylation and mutp53 protein levels resulted from FASN KO (**Supplementary Fig. 6d, e**). These results showed that extracellular palmitate has a mild and limited effect on mutp53 palmitoylation and mutp53 protein levels, which cannot fully rescue the decreased mutp53 palmitoylation and protein levels induced by FASN KO in cells. These new results also suggest that the palmitate produced via FASN-mediated de novo lipogenesis appears to play a more important role in regulating mutp53 palmitoylation and protein levels than exogenous palmitate that affects the general levels of palmitate in cells.

We totally agree with the Reviewer's comment that the palmitoylation status of proteins can be modulated by both palmitoylation and depalmitoylation processes in cells. As suggested, we investigated whether FASN affects the depalmitoylation process of mutp53 by using bioorthogonal labeling with the palmitate analog 17-octadecynoic acid (17-ODYA) followed by coupling to a fluorescent azide-reporter tag. Our results showed that FASN KO did not clearly affect the rate of depalmitoylation of mutp53 (**Supplementary Fig. 7a**). The depalmitoylating enzymes APT1 plays a critical role in protein depalmitoylation. Palmostatin B (Palm B), the small molecule APT1 inhibitor, was employed to treat cells to block the depalmitoylation of mutp53. While Palm B enhanced the palmitoylation of mutp53, notably, Palm B did not rescue the decreased mutp53 palmitoylation induced by FASN KO or TVB-3166 treatment (**Supplementary Fig. 7b, c**). This result suggests that FASN promotes mutp53 palmitoylation mainly through regulating the palmitoylation process instead of the depalmitoylation process of mutp53.

In sum, these results from our new experiments together with those results in our original submission suggest that FASN promotes mutp53 palmitoylation, and the direct interaction of FASN and mutp53 appears to play an important role in mutp53 palmitoylation.

This reviewer also raised a very good and important question regarding the FASN-mutp53 interaction as follows: *“The interaction of FASN with mutp53 over wild type p53 is strikingly selective as is the preferential palmitoylation of mutp53. While the authors narrow down the region of interaction to the DNA binding domain of mutp53 where both the GOF mutations and*

palmitoylated cysteines reside, it's unclear how these mutations mediate the novel interaction. The role of FASN in this process requires further investigation.” Reviewer 3 also raised this same question, so we have addressed this comment of this Reviewer in our response to Reviewer 3's comment 2. Please see our response to Reviewer 3's comment 2. Thanks!

Comment 3: The p53-FASN PLA assay shows the two proteins interact in the area of the nucleus, this should be quantified. Can the authors determine if the puncta are inside or outside the nuclear envelope? FASN is generally in the cytoplasm, a western blot of the nuclear and cytoplasmic fractions should be included to show the level of FASN in the nucleus. Nuclear FASN has been reported in prostate cancer (Madigan et al Am J Pathol. 2014 Aug;184(8):2156-62), but I am not aware if this is generally true in other cancer cells.

Response: Thanks for this great suggestion and for pointing out this important reference. Interestingly, it was reported that FASN exhibited nuclear localization in prostate cancer cells (Ref 34). As suggested, we quantified the PLA signals that represent FASN-mutp53 interaction and added the new data in **Fig. 1f** (lower right panel). The mutp53 and FASN interaction signals were detected in both cytoplasmic and nuclear compartments with more interaction signals detected in the nuclear compartment (**Fig. 1f**). We also performed cytoplasmic/nuclear fractionation followed by Western blot assays as suggested, and our results showed the presence of FASN in the nuclear fraction although more FASN proteins were in the cytoplasmic fraction in different cell lines, including SK-BR3, p53^{R248W/-} HCT116, MDA-MB468 and LS1034 cells (**Supplementary Fig. 1**). We have also cited this important reference in our manuscript (Ref 34, Page 7).

Comment 4: Twenty-four hours is a relatively long time to inhibit FASN and could have broad effects. A time course of pharmacologic inhibition of FASN may help clarify if the reduction in palmitoylation is caused by general depletion in palmitate (only after 24 hours) or a local decrease or if FASN directly palmitoylates mutp53 (short time points).

Does FASN inhibition for 24 hours reduce DHHC3 and 17 auto-palmitoylation (by ABE)? This would suggest that palmitate is limiting at this time point.

Response: Thanks for these two wonderful suggestions. As suggested, we conducted a time-course study for TVB-3166 treatment (0, 8, 16, and 24h) to monitor mutp53 palmitoylation, along with ZDHHC3/17 autopalmitoylation as a reference. Notably, TVB-3166 treatment induced a much faster decrease in mutp53 palmitoylation compared with ZDHHC3/17 palmitoylation (**Supplementary Fig. 6a**). At the same time, although at 24 h both ZDHHC3 and ZDHHC17 showed a mild reduction of autopalmitoylation, mutp53 showed a much more pronounced reduction of palmitoylation compared with ZDHHC3 and ZDHHC17 (**Supplementary Fig. 6a**). This result suggests that the reduction in palmitoylation of mutp53 appears to be caused by a local decrease of palmitate through the FASN-mediated de novo lipogenesis instead of the general depletion in palmitate in cells, which supports our hypothesis that FASN interacts with mutp53 to directly regulate mutp53 palmitoylation.

Comment 5: Does FASN inhibition prevent FASN interaction with mutp53? Conversely does FASN interact with mutp53 containing the cysteine mutations or when DHHC3/17 are knocked down? This would help determine if FASN binding requires the palmitoylated residues on mutp53 and binding is after palmitoylation.

Response: Thank you for this great suggestion. As suggested, we performed following experiments and new results are presented in this revised manuscript. As shown in **Supplementary Fig. 5**, TVB-3166 treatment did not clearly affect the mutp53-FASN interaction as analyzed by co-IP followed by Western blot assays. Furthermore, 4CS mutations that abolished the mutp53 palmitoylation did not clearly affect the interaction between mutp53 and FASN (**Supplementary Fig. 8b**). In addition, knockdown of ZDHHC3 and ZDHHC17 by siRNAs did not clearly affect the interaction between mutp53 and FASN (**Supplementary Fig. 11a**). These results suggest that palmitoylation status of mutp53 does not affect the interaction between FASN and mutp53.

Comment 6: Some molecular characterization of the treated and untreated tumors in addition to the p53 levels presented would strengthen the conclusions. It would be informative if the transcriptional changes known to occur in response to GOF mutp53 are altered in the tumors treated with the FASN inhibitors and if they are consistent with tumor growth inhibition. This doesn't need to be global analysis, just some of the known target genes. Is proliferation of the tumor cells reduced or is cell death increased?

Response: Thanks for this very insightful suggestion. As suggested, we investigated whether TVB-3166 treatment affects mutp53 GOF function in regulation of gene expression through its downregulation of mutp53 levels in tumors using real-time PCR assays. KO of R175H mutp53 in tumors significantly reduced the expression of several well-known mutp53-regulated genes, including ADRB1, IGFBP3 and P2RX5, confirming these genes are regulated by GOF mutp53 (**Supplementary Fig. 18**). Consistent with our results showing that TVB-3166 treatment greatly reduced mutp53 protein levels and palmitoylation in tumors (**Fig 6f**, and **Supplementary Fig. 19a**), TVB-3166 treatment significantly reduced the expression of these mutp53-regulated genes in tumors carrying mutp53, and furthermore, this effect of TVB-3166 was largely abolished in tumors with mutp53 KO, suggesting that TVB-3166 suppresses mutp53 GOF in regulation of gene expression (**Supplementary Fig. 18**).

As suggested, we also analyzed tumor cell proliferation by IHC staining of Ki-67 in tumors. Our results showed that TVB-3166 treatment significantly reduced Ki-67-positive cells in tumors carrying mutp53, and this effect of TVB-3166 was greatly reduced in tumors with mutp53 KO, suggesting that TVB-3166 treatment inhibits mutp53 GOF in promoting cell proliferation (**Fig 6h**). We also examined cell death in tumors treated with TVB-3166 by IHC staining of cleaved-caspase-3, and found that no significant difference was observed between tumors carrying mutp53 and tumors with mutp53 KO (data not shown). These results suggest that pharmacological inhibition of FASN suppresses mutp53 accumulation and GOF in promoting cancer cell proliferation and preferentially inhibits the growth of mutp53 tumors.

Minor comments.

Comment 7: I think general readers are less familiar with the gain of function mutations in p53. Can the authors elaborate on how frequent the GOF mutations are in different cancers?

Response: Thanks for this very good suggestion. p53 is the most frequently mutated gene in human cancers, and over 50% of all human cancers have p53 mutations. About 90% of these mutations encode missense mutant proteins that span ~190 different codons localized in the DNA-binding domain of the gene and protein. Six of these mutations are localized in codons that account for ~30% of the total p53 mutations in human cancers (R175, R248, R248, R273, R282, and R249), termed as mutation hotspots. While it was estimated that the majority of missense p53 mutations may have different levels of GOF activities (Ref 14), these six hotspot mutations have been shown to display GOF activities to promote tumorigenesis and widely used as representatives for GOF studies in numerous studies (Refs 3-13). We have added this introduction into our Introduction section (Page 3).

Comment 8: It would be informative as to why was FASN chosen for further study out of the proteomic interaction screen over other potential candidates.

Response: It is a great suggestion. Mutp53 has been suggested to be involved in regulation of metabolism, including lipid metabolism (Ref 31). However, its mechanism is not well understood. Furthermore, it has been well established that mutp53 protein is frequently stabilized and accumulated to display GOF activities in cancer, but its underlying mechanism is poorly understood. FASN not only plays a critical role in lipid metabolism but also palmitoylation modification, which can regulate the levels and functions of substrate proteins. Our co-IP followed by LC-MS/MS analysis suggests that FASN could be a novel and important binding protein for mutp53 (but not for wtp53), therefore, we decided to choose FASN and study its potential role and mechanism in the regulation of mutp53 protein accumulation and GOF. We have added this information in our Introduction section as suggested (page 5).

Reviewer #3's comments:

In the article 'Lipogenic enzyme FASN promotes mutant p53 accumulation and gain-of-function in cancer through palmitoylation' the authors characterize a role for FASN in preventing degradation of mutant p53 proteins. The article is well-written and logically structured. Experiments are well-controlled and data are interpretable.

Major comments:

Comment 1. In figure 3, the authors show that 4 cysteines in p53 are important for binding to FASN and palmitoylation of p53. Do the authors also see a reduced binding of ZDHHC3/17 to this mutant?

Response: Thanks very much for the kind comments and great suggestions!

Thanks for raising this very important question. As suggested, we examined whether mutations of these 4 cysteines in mutp53 (4CS mutations) affect the interactions between mutp53 and ZDHHC3/17 by co-IP assays, and found that the 4CS mutations did not affect the interaction between mutp53 and ZDHHC3/17 (**Supplementary Fig. 11b, c**).

Comment 2. One unresolved question is why FASN and the ZDDHCs bind various mutants, but not WT p53. The authors speculate in the discussion that this could be to do with the folding status of the mutants/ accessibility to p53. Some of the mutants that were analyzed are presumed to still normally fold. Does any other modification that occurs on cysteines in WT p53 prevent palmitoylation? Maybe glutathionylation? Or does DNA binding of Wtp53 prevent access to these cysteines.

Response: Thanks for raising this very important question. Missense mutp53 has been reported to specifically interact with some proteins to affect these proteins' functions, activities, and/or localizations (Refs 5, 7, 8, 13, 32). For instance, these 4 hotspot mutp53 that we tested in this study, including R175H, G245S, R248W and R273H, all interacted with FASN and ZDHHC3/17, although R175H mutp53 displayed stronger interactions with FASN and ZDHHC3/17 compared with the other 3 hotspot mutp53 (**Fig. 1a** and **Fig. 4f, g**). Currently, it is still unclear why mutp53 (but not wtp53) can specifically bind to these proteins. One possibility that has been proposed in the p53 research field is that missense mutations may affect the conformation of p53 protein, which leads to its interaction with some proteins different from wtp53. Currently, a significant amount of our knowledge on the mutp53 conformation changes came from the studies using antibodies against different forms of mutp53 (Ref 58). And the classification of two major categories of mutp53, conformational mutants (e.g. R175H and G245S) and contact mutants (e.g. R248W and R273H), also came from the use of antibodies (Ref 58). Interestingly, a recent study investigated the structural and dynamic properties of R175H and R273H mutp53, two representatives of two major categories of mutp53, by performing extensive all-atom molecular dynamics simulations and revealed distinct conformations for both R175H and R273H mutants compared to wtp53 (Ref 59). This study suggests that many different forms of mutp53 may also have conformation changes although they belong to contact mutants, which may affect their interactions with other proteins.

We totally agree with the Reviewer's comment that post-translation modifications of mutp53 may affect mutp53 protein conformation and the accessibility of other proteins or enzymes to mutp53. Given that many different sites in p53 that can undergo different types of modifications, it has been suggested that there is crosstalk between these modifications, which can impact the modifications, conformations and functions of p53 protein (Ref 60). For the case of mutp53 palmitoylation, other types of post-translation modifications of mutp53 may affect the accessibility of FASN and/or ZDHHC3/17 to mutp53. We thank the Reviewer for highlighting the potential connection between palmitoylation and glutathionylation of p53. It was reported that the DNA binding function of wtp53 can be inhibited by glutathionylation at cysteines (C124, C141, and C182) under oxidative stress (Ref 62). It is possible that glutathionylation at cysteines may inhibit the palmitoylation modification of wtp53 and/or mutp53. We also agree with the Reviewer that the binding of wtp53 to DNA may prevent the access of FASN and ZDHHC3/17 to these cysteines. We appreciate the Reviewer raising these very important questions and giving very good suggestions for future

studies. Obviously, many more studies should be done in future to address these important questions in the p53 research field. We have added these discussions into our Discussion section (Pages 22, 23).

Comment 3. It would be good if the authors could put the current findings in the context of other literature around p53 and FASN. E.g. Li et al 2001 PMID: 11245456 and D-'Erchia 2006 PMID: 16582625. Especially the first article already describes the usefulness of FASN inhibitors in cells that have no functional p53.

Response: Thanks for pointing out these two important articles. As suggested, we have cited these two references and incorporated this important information into our Discussion section as follows (Refs 56, 57; Pages 21, 22).

“In line with our results, a previous study showed that cell lines lacking p53 or expressing mutp53 exhibit increased sensitivity to apoptosis induced by FASN inhibition (Ref 56). Interestingly, it was also reported that FASN in *C. elegans* is a target gene of CEP-1, the p53 homolog in *C. elegans*, and the human FASN gene is a target of p53 family members TAp73 α and Δ Np63 α but not of p53, which suggests an interesting but complex link between FASN and members of the p53 family (Ref 57).”

Comment 4. Do the authors see a difference in numbers of tumours/aggressiveness of the tumours in the mice used in figure 6K? Do tumours in the TVB treated mice have lower palmitoylation and lower 172H levels?

Response: Thanks for raising these two great questions.

We did not see a clear difference in numbers of tumors/aggressiveness of the tumors in the mice used in **Fig. 6k** (**Fig. 6l** in this revised manuscript) as shown in **Supplementary Table 2**.

Consistent with our observation that TVB-3166 treatment reduced mutp53 protein levels in SK-BR3 orthotopic breast tumors (collected one day after the completion of TVB-3166 treatment) (**Fig. 6f**), TVB-3166 also drastically reduced mutp53 palmitoylation in these tumors (**Supplementary Fig. 19a**). However, in the SK-BR3 tumors collected 10 days after the completion of TVB-3166 treatment, the decrease in mutp53 palmitoylation and protein levels largely disappeared compared with control tumors without TVB-3166 treatment (**Supplementary Fig. 19b, c**), indicating that the effect of FASN inhibition on mutp53 palmitoylation and protein levels in tumors gradually disappeared after TVB-3166 withdrawal. It is worth mentioning that majority of these p53^{R172H/R172H} mice succumbed to mortality far more than 10 days after the completion of TVB-3166 treatment (for 12 weeks) when their tumors were collected and analyzed (**Fig. 6l**). Consistent with our observation that the effect of FASN inhibition by TVB-3166 on mutp53 palmitoylation and protein levels in SK-BR3 tumors gradually disappeared after TVB-3166 withdrawal (**Supplementary Fig. 19a-c**), there was no clear difference in mutp53 palmitoylation levels between the tumors collected from p53^{R172H/R172H} mice treated with TVB-3166 and tumors from p53^{R172H/R172H} mice treated with vehicle only (**Supplementary Fig. 19d**).

Reviewer #4's comments

Liu et al, present an interesting study on the occurrence and function of palmitoylation in the gain-of-function(GOF) mutant of p53. The authors suggest that the p53GOF mutant physically interacts with FASN and undergoes palmitoylation catalyzed by ZDHHC3 and ZDHHC17. While the palmitoylation-mediated regulation of wild-type p53 has been documented before, this study highlights distinctions in the mechanism and functional consequences of palmitoylation in the context of the p53 GOF mutation. This work is clearly explained, and the experiments, conducted across various cell lines, seem robust and their outcome convincing. Nonetheless, there are a number of points that require attention.

Comment 1. It remains unclear why and how all distinct GOF mutations facilitate binding to FASN, ZDHHC3, and ZDHHC17, leading to palmitoylation, whereas this phenomenon does not occur in wild-type (wt) p53. Do various GOF mutations induce comparable structural changes to the protein? Do the diverse p53 GOF mutants elicit comparable alterations to the cellular state? What determines the differential binding and regulation of palmitoylation among these mutants?

Response: Thanks very much for the kind comments and great suggestions!

Thanks for raising these very important questions. Missense mutp53 has been reported to specifically interact with many proteins that do not bind to wtp53, and affect these proteins' functions, activities, and/or localizations (Refs 5, 7, 8, 13, 32). At the same time, different forms of mutp53 have been reported to differ in the magnitudes of their GOF activities and display GOF activities in a highly cell- or tissue-specific and context-dependent manner (Refs 5, 54, 55). Different forms of mutp53 have also been shown to interact with some different proteins although they can interact with some common proteins. Interestingly, although these 4 hotspot mutp53 that we tested in this study, including R175H, G245S, R248W and R273H, all interacted with FASN and ZDHHC3/17, R175H mutp53 displayed stronger interactions with FASN and ZDHHC3/17 compared with the other 3 hotspot mutp53 (**Fig. 1a** and **Fig. 4f, g**). It is still unclear why mutp53 (but not wtp53) can specifically bind to these proteins. One possibility has been proposed in the p53 research field is that missense mutations may affect the conformation of p53 protein and the conformation of different mutp53 may be differently affected, which leads to the interactions of mutp53 with some specific proteins different from wtp53, such as FASN in this study. For example, a recent study investigated the structural and dynamic properties of R175H and R273H mutp53 by performing extensive all-atom molecular dynamics simulations. Their findings revealed distinct conformations for both R175H and R273H mutants compared to wtp53 (Ref 59).

Since Reviewer 3 also raised similar questions, which we have addressed, more discussions on these questions can be found in our response to Reviewer 3's comment 2. We totally agree with the Reviewer that these important questions have not been well addressed in the field, which should be addressed by future studies. We appreciate the Reviewer raising these important questions and have added this discussion into our Discussion section (Pages 22, 23).

Comment 2. Building on the previous point, do p53 gain-of-function (GOF) mutations establish a comparable cellular environment that facilitates or promotes the palmitoylation process? This question could be addressed through comparative analyses of RNAseq, proteomics, and/or metabolomics.

Response: Thanks for raising this very important question. Currently, it remains unclear how mutp53 specifically interacts with FASN and ZDHHC3/17 (while wtp53 does not) and establishes a cellular environment to promote the palmitoylation process. It has been reported that GOF mutp53 often exhibits altered interactions with other proteins, dysregulated gene expression patterns, and aberrant signaling pathways in cancer cells, which collectively contribute to its GOF in tumorigenesis. Further, different forms of mutp53 may differ in the magnitudes of their GOF activities and display GOF activities in a highly cell- or tissue-specific and context-dependent manner. It is a great suggestion that future studies using comprehensive strategies, including comparative analyses of RNAseq, proteomics, and/or metabolomics could shed further light on this important question. We have incorporated this discussion into the Discussion section (Pages 21, 23).

Comment 3. What is the contribution of variable levels of palmitate to WT and GOF p53 palmitoylation? Despite previous descriptions of FASN acetyltransferase activity, the current study reveals that knockdown of ZDHHC3 and ZDHHC17 prevents palmitoylation of p53 mutants, suggesting that FASN does not fulfill this role. Can elevated palmitate levels overcome the absence of FASN? The authors should also discuss the potential impact of factors such as, for instance a high-fat diet, in the regulation of WT and GOF mutated p53 in cancer.

Response: Thanks for these great suggestions. We have done following experiments as suggested. Since palmitate (PA) can trigger toxicity and cell death in cells at high doses, we tested different concentrations of palmitate in cells and observed reduced cell viability after 24 h at concentrations exceeding 60 μ M. Therefore, we treated cells with 30 and 60 μ M palmitate (for 24 h), and found that palmitate treatment resulted in a slight increase in the palmitoylation of both mutp53 and wtp53 in cells (**Supplementary Fig. 6b**). Similarly, palmitate treatment slightly but not fully rescued the decreased mutp53 palmitoylation and mutp53 protein levels resulted from FASN KO (**Supplementary Fig. 6c**). Consistent with our results showing that palmitoylation did not affect wtp53 protein levels in cells (**Fig. 2**), palmitate treatment did not clearly affect the wtp53 protein levels in cells (**Supplementary Fig. 6d, e**). Together, these results indicate that extracellular palmitate has a mild and limited effect on mutp53 palmitoylation and protein levels and cannot fully rescue the decreased mutp53 palmitoylation and protein levels induced by FASN KO in cells. These results also suggest that palmitate produced via FASN-mediated de novo lipogenesis might be critical for mutp53 palmitoylation and protein levels.

Currently, it is unclear whether dietary fatty acid supplementation or a high-fat diet can effectively rescue the decreased mutp53 palmitoylation and impaired GOF in tumorigenesis resulted from FASN deficiency or inhibition in animal models. Interestingly, several recent studies reported the dietary fatty acid supplementation or high-fat diet alone cannot effectively reverse the phenotypes resulted from FASN inhibition in mouse models. For instance, exogenous palmitate supplementation failed to prevent vision defects resulted from FASN KO in mice, suggesting the

de novo lipogenesis-dependent pathways intrinsic to the eye are crucial for vision (Ref 63). FASN was reported to be crucial for accurate myelination and myelin growth, and dietary lipid intake only partially compensated the defects caused by FASN deficiency in mice (Ref 64). Additionally, FASN deletion in macrophages prevented diet-induced insulin resistance, macrophage recruitment to adipose tissue, and chronic inflammation in mice, and exogenous palmitate failed to rescue these phenotypes caused by FASN deletion (Ref 65). These findings suggest that locally and newly synthesized palmitate through the FASN-mediated de novo lipogenesis other than exogenous palmitate from extracellular sources plays a crucial role in regulation of some critical cellular functions (Refs 63, 66). Based on our results observed in cultured cells (**Supplementary Fig. 6b-e**) and these previous studies, the dietary fatty acid supplementation or a high-fat diet may partially but not fully rescue the decreased mutp53 palmitoylation and impaired GOF in tumorigenesis resulted from FASN deficiency or inhibition in animal models, which deserves further studies in future. We have incorporated these references and discussion into the Discussion section (Pages 23-24).

Comment 4. FASN deletion has, as expected a strong effect on tumour development/growth. Although the effect seems more pronounced in GOFp53 tumours, it is also the case that those tumours grow more/faster which entitles larger demands of fatty acids and other (anabolic) metabolites. Therefore, it is misleading to interpret that the primary function of FASN is mediated through GOF p53 palmitoylation. Can the deletion of FASN, along with the simultaneous overexpression of stabilized p53, rescue tumor or organoid growth and development?

Response: Thanks for raising this important point and for this great suggestion. We totally agree with the reviewer that it has been well-established that FASN overexpression and/or activation plays an important role in promoting tumorigenesis through supplying fatty acids for membrane synthesis and lipid metabolism in cancer cells. We have revised our description in the Results part to avoid the misleading and misunderstanding.

As suggested, we further investigated whether restoration of the expression of mutp53 in cells with FASN KO can rescue the impaired Anchorage-independent growth (AIG) of cells on soft agar and tumorigenesis resulted from FASN KO. SK-BR3 cells with FASN KO were stably transduced with the R175H mutp53 expression vector to increase the level of mutp53 protein to a comparable level of endogenous mutp53 in control SK-BR3 cells without FASN KO. Cells were then employed for AIG and tumorigenesis assays. Restoration of the expression of R175H mutp53 in cells with FASN KO significantly but did not fully rescue the impaired AIG and tumorigenesis resulted from FASN KO (**Supplementary Fig. 15 a-c**), suggesting that while the palmitoylation of mutp53 by FASN contributes greatly to the promoting effects of FASN on AIG and tumorigenesis, the known functions of FASN, especially lipid synthesis, also play important roles in promoting AIG and tumorigenesis. We have added new results and discussion (**Supplementary Fig. 15**).

Minor

Comment 5. Regarding the title:

'Lipogenic Enzyme FASN Promotes Mutant p53 Accumulation and Gain-of-function in Cancer through Palmitoylation' Consider to remove 'in cancer' as the above seems also to occurs in wt cells (MEFs) if p53mut GOF is introduced?

Response: As suggested, we have removed “in cancer” from the title. Thanks!

Comment 6. Table 1: Potential mutp53-interacting proteins identified by co-IP followed by LC-MS/MS analysis. All interactions are higher for mutp53 when compared to wt p53. How was the data corrected/normalized? Were comparable levels of p53 pulled down? Did bona fide interactors of wt and GOF p53 appeared in the pulldowns? Do different GOF mutants present comparable interactomes (related to point 1).

Response: Yes, comparable levels of p53 were pulled down confirmed by Western blot analysis in both wtp53 and mutp53 samples to ensure that any differences in interaction levels were not due to the difference in p53 abundance but rather due to differences in the interactions with other proteins. Mutp53 and wtp53 exhibit comparable total spectral counts. Different GOF mutp53 may not necessarily present comparable interactomes, but some interactors are common across different GOF mutants. For example, HSP90AB1, HSPA1A, HSPA8, and BAG2 as listed in the Supplementary Table 1 have been reported to interact with different GOF mutp53, including R175, R273, R248, etc (Refs 5, 8, 13, 32). In this study, we also found that FASN can bind with different hotspot mutp53, including R175H, G245S, R248W and R273H, although its binding to R175H mutp53 appears to be stronger compared with the other 3 hotspot mutp53 (**Fig. 1a**).

Comment 7. The quality of fluorescent images across the manuscript seems to be low and oversaturated.

Response: Thanks! As suggested, we have replaced them with images with better quality.

Again, we really appreciate you all for carefully reviewing our manuscript, and for the nice and very insightful comments, very constructive and helpful suggestions, and great efforts to improve our manuscript. We wish that with our changes and response, our manuscript could be acceptable for publication.

Thank you very much!

Sincerely yours,

Zhaohui

Zhaohui Feng, MD, PhD

Professor

Rutgers Cancer Institute of New Jersey

Rutgers, The State University of New Jersey

195 Little Albany Street, New Brunswick, NJ 08903-2681, USA

Email: fengzh@cinj.rutgers.edu

Phone: 732-690-7296

Point-by-point response to Reviewers' comments

First, we would like to thank the four Expert Reviewers for taking precious time to review our revised manuscript and providing positive, very insightful and constructive comments on our manuscript. We have carefully revised our manuscript accordingly and appropriate changes have been incorporated into our manuscript. Following is our response to the comments from reviewers.

REVIEWER COMMENTS

Reviewer #1 (Remarks to the Author):

*Comments: The authors have addressed the concerns raised adequately and this is reflected in the significantly improved manuscript.
I recommend the manuscript for publication.*

Response: Thanks very much for the very kind comments!

Reviewer #2 (Remarks to the Author):

Comments: The authors have addressed most of my concerns and have ruled out that FASN is preventing depalmitoylation of mutp53 and confirmed the presence of FASN in the nucleus of the cell lines used. They have quantified the PLA study and clarified that the interaction between FASN and mutp53 is predominately in the nucleus. They have confirmed that DHHC mediated palmitoylation of mutp53 does not mediate the interaction with FASN. They have shown ZDHHC3/17 interacts with mutp53 in the nucleus by PLA and by western blotting shown that ZDHHC3/17 is in the nucleus.

The authors removed the statement that FASN transfers palmitate to mutp53 and appear to favor the mechanism where FASN provides palmitate through de novo lipogenesis for DHHC mediated transfer to mutp53. They have presented strong evidence for this mechanism even though exogenous palmitate only minimally rescued mutp53 palmitoylation in the FASN KO cells.

The author states: These results showed that extracellular palmitate has a mild and limited effect on mutp53 palmitoylation and mutp53 protein levels, which cannot fully rescue the decreased mutp53 palmitoylation and protein levels induced by FASN KO in cells. These new results also suggest that the palmitate produced via FASN-mediated de novo lipogenesis appears to play a more important role in regulating mutp53 palmitoylation and protein levels than exogenous palmitate that affects the general levels of palmitate in cells.

Response: Thanks very much for the very kind comments!

Comments: It is important to note that the product of FASN is palmitoyl-CoA the substrate for the DHHC enzymes, not palmitate and this should be added to the schematic in Fig. 7F. Palmitate needs to be converted to palmitoyl-CoA when provided exogenously and could explain the weak effect on rescue. It is possible that palmitoyl-CoA is low in the nucleus and resides in other membrane compartments. However, this should also be true for the click analog 17-ODYA used for labeling.

Response: Thanks for the great suggestion! As suggested, we have revised our schematic in Fig. 7F by adding palmitoyl-CoA as the substrate for the DHHC enzymes.

Comments: Why is the incorporation of 17-ODYA into mutp53 after 2 hours incubation equivalent between control HECT116 and HECT116 FASN KO cells in Supp Fig 7a while exogenous palmitate treatment after 24 hours in the SK-BR3 (R175H) FASN KO cells have drastically reduced palmitoylation compared to control cells? If exogenous palmitate is less efficient at palmitoylating mutp53 than de novo lipogenesis I would expect a similar reduction in labeling with the 17-ODYA. Did the authors try the palmitate rescue experiment in the HECT116 cells? It might clarify what is happening with the exogenous palmitate experiment if these experiments were performed in the same cell lines. Is there a difference in the cell culture conditions between the click experiment and the palmitate rescue that could explain this difference? This should be clarified before publication especially if this is caused by a cell line specific effect related to conversion of palmitate to palmitoyl-CoA.

Response: Thanks for raising this excellent question! We apologize for causing the confusion regarding our results in Supplementary Fig. 7a. shown in our previous version of manuscript.

Indeed, we observed that the incorporation of 17-ODYA into mutp53 after 2 hours incubation with 17-ODYA was much higher in control HCT116 cells than HCT116 FASN KO cells, which is consistent with the reviewer's prediction and also consistent with results in Supplementary Fig. 6c showing that exogenous palmitate treatment in the SK-BR3 (R175H) FASN KO cells led to the drastically reduced palmitoylation levels of mutp53 compared with control SK-BR3 cells.

In the Supplementary Fig. 7a. in our previous version of manuscript (as shown in the upper figure in this letter), we presented the Western-blot results of HCT116 and HCT116 FASN KO cells on two different gels with different exposure times. The reason to present in this way was trying to make the incorporation levels of 17-ODYA into mutp53 in the two cell lines comparable, thereby facilitating a clear comparison of the rate of mutp53 depalmitoylation between the two cell lines (as shown in the right panels: quantification of palmitoylated mutp53). However, this presentation caused confusion regarding the levels of 17-ODYA incorporation into mutp53 after 2 hours of incubation in the two cell lines. We appreciate the reviewer for pointing this out.

As suggested, we have revised Supplementary Fig. 7a. (shown in the lower figure in this letter) by presenting the Western-blot results from two cell lines on the same gel with both short and long exposures, which clearly shows the drastically reduced levels of mutp53 palmitoylation in HCT116 FASN KO cells compared with control HCT116 cells, while also allowing for a clear comparison of the rate of mutp53 depalmitoylation between the two cell lines.

Again, we greatly appreciate the reviewer for raising this important question!

The original Supplementary Fig. 7a

The revised Supplementary Fig. 7a

Reviewer #3 (Remarks to the Author):

Comment: The authors have very thoroughly and successfully addressed all my concerns.

Response: Thanks very much for the kind comment!

Reviewer #4 (Remarks to the Author):

Comment: The authors have clearly addressed the points I raised. Thanks!

Response: Thanks very much for the very kind comment!

Minor:

Supplementary Table 1: Potential mutp53-interacting proteins identified by co-IP followed by LC-MS/MS analysis.

Please extend the table to all found proteins in the pulldowns, and upload processed and raw data to public repositories.

Response: Thanks for this great suggestion! As suggested, we have extended the Supplementary Table 1 to all found proteins in the pulldowns. However, we encountered some challenges in obtaining the raw data from our co-facilities in order to deposit the protein interaction data to public repositories.

Our lab has been working on mutant p53 (mutp53) and mutp53-interacting proteins since I started my own lab 15 years ago. The mutp53-interacting proteins analyzed by co-IP followed by LC-MS/MS analysis in HCT116 cells as we described in this manuscript was done in 2012 by the co-facility of our Rutgers University. Since majority of journals do not request the raw data of protein co-IP followed by LC-MS/MS analysis to be deposited in public repositories, the report from our facility only contains processed data but not raw data. After we received the comment from the reviewer, we contacted the co-facility many times and also personally met with the technicians and manager in the co-facility several times. They put in great effort and tried very hard to look through their computers to look for the raw data of our analysis. Unfortunately, they can't find the raw data since almost 12 years passed and their computer systems have been updated many times. Without the raw data, we are not allowed to deposit processed data only to the public repositories. As suggested, we have now provided the processed data in the Supplementary Table 1 to allow readers have the access to this information. We sincerely apologize for any inconvenience caused by this issue!

Again, we really appreciate four expert reviewers for carefully reviewing our manuscript, and for the very nice and insightful comments, very constructive and helpful suggestions, and great efforts to improve our manuscript. We wish that with our changes and response, our manuscript could be acceptable for publication.

Thank you very much!

Sincerely yours,

Zhaohui

Zhaohui Feng, MD, PhD

Professor

Rutgers Cancer Institute

Rutgers, The State University of New Jersey
195 Little Albany Street, New Brunswick, NJ 08903-2681, USA
Email: fengzh@cinj.rutgers.edu
Phone: 732-690-7296

Point-by-point response to Reviewers' comments

First, we would like to thank the four Expert Reviewers for taking precious time to review our revised manuscript and providing positive, very insightful and constructive comments on our manuscript. We have carefully revised our manuscript accordingly and appropriate changes have been incorporated into our manuscript. Following is our response to the comments from reviewers.

REVIEWER COMMENTS

Reviewer #1 (Remarks to the Author):

*Comments: The authors have addressed the concerns raised adequately and this is reflected in the significantly improved manuscript.
I recommend the manuscript for publication.*

Response: Thanks very much for the very kind comments!

Reviewer #2 (Remarks to the Author):

Comments: The authors have addressed most of my concerns and have ruled out that FASN is preventing depalmitoylation of mutp53 and confirmed the presence of FASN in the nucleus of the cell lines used. They have quantified the PLA study and clarified that the interaction between FASN and mutp53 is predominately in the nucleus. They have confirmed that DHHC mediated palmitoylation of mutp53 does not mediate the interaction with FASN. They have shown ZDHHC3/17 interacts with mutp53 in the nucleus by PLA and by western blotting shown that ZDHHC3/17 is in the nucleus.

The authors removed the statement that FASN transfers palmitate to mutp53 and appear to favor the mechanism where FASN provides palmitate through de novo lipogenesis for DHHC mediated transfer to mutp53. They have presented strong evidence for this mechanism even though exogenous palmitate only minimally rescued mutp53 palmitoylation in the FASN KO cells.

The author states: These results showed that extracellular palmitate has a mild and limited effect on mutp53 palmitoylation and mutp53 protein levels, which cannot fully rescue the decreased mutp53 palmitoylation and protein levels induced by FASN KO in cells. These new results also suggest that the palmitate produced via FASN-mediated de novo lipogenesis appears to play a more important role in regulating mutp53 palmitoylation and protein levels than exogenous palmitate that affects the general levels of palmitate in cells.

Response: Thanks very much for the very kind comments!

Comments: It is important to note that the product of FASN is palmitoyl-CoA the substrate for the DHHC enzymes, not palmitate and this should be added to the schematic in Fig. 7F. Palmitate needs to be converted to palmitoyl-CoA when provided exogenously and could explain the weak effect on rescue. It is possible that palmitoyl-CoA is low in the nucleus and resides in other membrane compartments. However, this should also be true for the click analog 17-ODYA used for labeling.

Response: Thanks for the great suggestion! As suggested, we have revised our schematic in Fig. 7F by adding palmitoyl-CoA as the substrate for the DHHC enzymes.

Comments: Why is the incorporation of 17-ODYA into mutp53 after 2 hours incubation equivalent between control HECT116 and HECT116 FASN KO cells in Supp Fig 7a while exogenous palmitate treatment after 24hours in the SK-BR3 (R175H) FASN KO cells have drastically reduced palmitoylation compared to control cells? If exogenous palmitate is less efficient at palmitoylating mutp53 than de novo lipogenesis I would expect a similar reduction in labeling with the 17-ODYA. Did the authors try the palmitate rescue experiment in the HECT116 cells? It might clarify what is happening with the exogenous palmitate experiment if these experiments were performed in the same cell lines. Is there a difference in the cell culture conditions between the click experiment and the palmitate rescue that could explain this difference? This should be clarified before publication especially if this is caused by a cell line specific effect related to conversion of palmitate to palmitoyl-CoA.

Response: Thanks for raising this excellent question! We apologize for causing the confusion regarding our results in Supplementary Fig. 7a. shown in our previous version of manuscript.

Indeed, we observed that the incorporation of 17-ODYA into mutp53 after 2 hours incubation with 17-ODYA was much higher in control HCT116 cells than HCT116 FASN KO cells, which is consistent with the reviewer's prediction and also consistent with results in Supplementary Fig. 6c showing that exogenous palmitate treatment in the SK-BR3 (R175H) FASN KO cells led to the drastically reduced palmitoylation levels of mutp53 compared with control SK-BR3 cells.

In the Supplementary Fig. 7a. in our previous version of manuscript (as shown in the upper figure in this letter), we presented the Western-blot results of HCT116 and HCT116 FASN KO cells on two different gels with different exposure times. The reason to present in this way was trying to make the incorporation levels of 17-ODYA into mutp53 in the two cell lines comparable, thereby facilitating a clear comparison of the rate of mutp53 depalmitoylation between the two cell lines (as shown in the right panels: quantification of palmitoylated mutp53). However, this presentation caused confusion regarding the levels of 17-ODYA incorporation into mutp53 after 2 hours of incubation in the two cell lines. We appreciate the reviewer for pointing this out.

As suggested, we have revised Supplementary Fig. 7a. (shown in the lower figure in this letter) by presenting the Western-blot results from two cell lines on the same gel with both short and long exposures, which clearly shows the drastically reduced levels of mutp53 palmitoylation in HCT116 FASN KO cells compared with control HCT116 cells, while also allowing for a clear comparison of the rate of mutp53 depalmitoylation between the two cell lines.

Again, we greatly appreciate the reviewer for raising this important question!

The original Supplementary Fig. 7a

The revised Supplementary Fig. 7a

Reviewer #3 (Remarks to the Author):

Comment: The authors have very thoroughly and successfully addressed all my concerns.

Response: Thanks very much for the kind comment!

Reviewer #4 (Remarks to the Author):

Comment: The authors have clearly addressed the points I raised. Thanks!

Response: Thanks very much for the very kind comment!

Minor:

Supplementary Table 1: Potential mutp53-interacting proteins identified by co-IP followed by LC-MS/MS analysis.

Please extend the table to all found proteins in the pulldowns, and upload processed and raw data to public repositories.

Response: Thanks for this great suggestion! We have revised our manuscript as suggested.

There is one situation that we would like to mention. The original experiment of the co-IP followed by LC-MS/MS analysis of the mutp53-interacting proteins in HCT116 cells as we described in our last submission was done almost 10 years ago by the co-facility of our Rutgers University. Routinely, our facility only provides processed data but not raw data to us. After we received the comment from the reviewer, we contacted the co-facility several times and also personally met with the technicians and manager in the co-facility several times. They put in great effort and tried very hard to look through their computers to look for the raw data of our analysis. Unfortunately, they can't find the raw data since their computer systems have been updated many times during the past 10 years.

Therefore, we discussed with the editor, and we were suggested to repeat the experiment of the co-IP followed by LC-MS/MS analysis in HCT116 cells so that we can deposit raw data in public repositories. As suggested, we have repeated our LC-MS/MS experiments to analyze mutant p53 interacting proteins in HCT116 cells, and very similar results were obtained as our previous LC-MS/MS experiments presented in our last submission. As shown in our revised Supplementary Table 1, our new results from LC-MS/MS experiments showed that FASN specifically binds to mutant p53 but not wild-type p53 in cells. The raw data of LC-MS/MS experiments has been deposited in MassIVE Repository (<https://massive.ucsd.edu/ProteoSAFe/static/massive.jsp>) (accession number: MSV000095580). Please see our raw data using the following link.

<https://nam02.safelinks.protection.outlook.com/?url=http%3A%2F%2Fmassive.ucsd.edu%2FProteoSAFe%2Fstatus.jsp%3Ftask%3Dc8ca7b4ba93648c3abef80e1a369a4b9&data=05%7C02%7CHaiyanz%40cabm.rutgers.edu%7C4761bf73d85c4e90164f08dcb049525%7Cb92d2b234d35447093ff69aca6632ffe%7C1%7C0%7C638590875574600681%7CUnknown%7CTWFpbGZsb3d8eyJWIjoiMC4wLjAwMDAiLCJQIjoiV2luMzIiLCJBTiI6IjEhaWwiLCJXVCI6Mn0%3D%7C0%7C%7C%7C&sdata=AoLi63xDAF5HljAPO3Wwnl3q11VNaT9ADewEiUwTxs%3D&reserved=0>

We have extended our Supplementary Table 1 of the mutant p53-interacting proteins, and also provided the processed data in our Source Files as suggested. Again, thanks for this great suggestion!

Again, we really appreciate four expert reviewers for carefully reviewing our manuscript, and for the very nice and insightful comments, very constructive and helpful suggestions, and great efforts to improve our manuscript. We wish that with our changes and response, our manuscript could be acceptable for publication.

Thank you very much!

Sincerely yours,

Zhaohui

Zhaohui Feng, MD, PhD

Professor

Rutgers Cancer Institute

Rutgers, The State University of New Jersey

195 Little Albany Street, New Brunswick, NJ 08903-2681, USA

Email: fengzh@cinj.rutgers.edu

Phone: 732-690-7296

Point-by-point response to Reviewers' comments

First, we would like to thank the Reviewer #2 for taking precious time to review our revised manuscript and providing very insightful and constructive comments on our manuscript. We have carefully revised our manuscript accordingly and appropriate changes have been incorporated into our manuscript. Following is our response to the comments from reviewer #2.

REVIEWER COMMENTS

Reviewer #2 (Remarks to the Author):

"The authors have addressed all of my concerns. Thanks!"

With regards to the authors addressing the request by another reviewer to present an longer list of MS identified proteins.

The overall list looks good, but the table provided is shorter than the excel spreadsheet and they should indicate how the list was curated. It looks like it is only proteins that are enriched with the mutant p53. The link to access the MS data does not work for me and I was unable to review it."

Response:

Thanks for the insightful observation! Yes, we only listed the proteins that are enriched with the mutant p53 (mutp53) in our original Supplementary Table 1 since we wanted to focus on the potential mutp53-interacting proteins only. WT p53 was used as a control.

In this revised manuscript, according to the Editors' suggestion, we have prepared a concise Excel file, as Supplementary Data 1 (the original Supplementary Table 1) entitled "The potential mutp53-interacting proteins identified by co-IP followed by LC-MS/MS analysis". This Supplementary Data 1 includes only the proteins specifically enriched in the mutp53 pull-down group compared to the control group, focusing on those with spectral counts ≥ 4 . The full list of the potential mutp53- and WT p53-interacting proteins has been provided in our Source Data.

We have provided the detailed instructions for accessing our deposited LC-MS data in the public depository in our last response to the editor's email.

Again, we really appreciate four expert reviewers for carefully reviewing our manuscript, and for the very nice and insightful comments, very constructive and helpful suggestions, and great efforts to improve our manuscript.

Thank you very much!

Sincerely yours,

Zhaohui

Zhaohui Feng, MD, PhD

Professor

Rutgers Cancer Institute

Rutgers, The State University of New Jersey

195 Little Albany Street, New Brunswick,

NJ 08903-2681, USA

Email: fengzh@cinj.rutgers.edu

Phone: 732-690-7296